# Reinforcement Learning with Fast and Forgetful Memory

**Steven Morad**
University of Cambridge
Cambridge, UK
sm2558@cam.ac.uk

**Ryan Kortvelesy**
University of Cambridge
Cambridge, UK
rk627@cam.ac.uk

**Stephan Liwicki**
Toshiba Europe Ltd.
Cambridge, UK
stephan.liwicki@toshiba.eu

**Amanda Prorok**
University of Cambridge
Cambridge, UK
asp45@cam.ac.uk

## Abstract

Nearly all real world tasks are inherently partially observable, necessitating the use of memory in Reinforcement Learning (RL). Most model-free approaches summarize the trajectory into a latent Markov state using memory models borrowed from Supervised Learning (SL), even though RL tends to exhibit different training and efficiency characteristics. Addressing this discrepancy, we introduce Fast and Forgetful Memory, an algorithm-agnostic memory model designed specifically for RL. Our approach constrains the model search space via strong structural priors inspired by computational psychology. It is a drop-in replacement for recurrent neural networks (RNNs) in recurrent RL algorithms, achieving greater reward than RNNs across various recurrent benchmarks and algorithms *without changing any hyperparameters*. Moreover, Fast and Forgetful Memory exhibits training speeds two orders of magnitude faster than RNNs, attributed to its logarithmic time and linear space complexity. Our implementation is available at https://github.com/proroklab/ffm.

## 1 Introduction

Reinforcement Learning (RL) was originally designed to solve Markov Decision Processes (MDPs) [Sutton and Barto, 2018], but many real world tasks violate the Markov property, confining superhuman RL agents to simulators. When MDPs produce noisy or incomplete observations, they become Partially Observable MDPs (POMDPs). Using *memory*, we can summarize the trajectory into a Markov state estimate, extending convergence guarantees from traditional RL approaches to POMDPs [Kaelbling et al., 1998].

In model-free RL, there are two main approaches to modeling memory: (1) RL-specific architectures that explicitly model a probabilistic belief over Markov states [Kaelbling et al., 1998] and (2) general-purpose models such as RNNs or transformers that distill the trajectory into a fixed-size latent Markov state. Ni et al. [2022] and Yang and Nguyen [2021] reveal that with suitable hyperparameters, general-purpose memory often outperforms more specialized belief-based memory.

Most applications of memory to RL tend to follow Hausknecht and Stone [2015], using RNNs like Long Short-Term Memory (LSTM) or the Gated Recurrent Unit (GRU) to summarize the trajectory, with other works evaluating transformers, and finding them tricky and data-hungry to train [Parisotto et al., 2020]. Morad et al. [2023] evaluate a large collection of recent memory models across many

partially observable tasks, finding that the GRU outperformed all other models, including newer models like linear transformers. Interestingly, they show that there is little correlation between how well memory models perform in SL and RL. Even recent, high-profile RL work like Hafner et al. [2023], Kapturowski et al. [2023] use RNNs over more modern alternatives, raising the question question: why do older memory models continue to overshadow their contemporary counterparts in modern RL?

Training model-free RL policies today is sample inefficient, alchemical, and prone to collapse [Schulman et al., 2015, Zhang et al., 2018], and reasoning over the entire trajectory magnifies these issues. SL can better utilize scale, compute, and dataset advancements than RL, largely removing the need for strong *inductive biases*. For example, transformers execute pairwise comparisons across all inputs, in a somewhat "brute force" approach to sequence learning. Similarly, State Space Models [Gu et al., 2021] or the Legendre Memory Unit [Voelker et al., 2019] are designed to retain information over tens or hundreds of thousands of timesteps, growing the model search space with each additional observation.

Through strong inductive biases, older recurrent memory models provide sample efficiency and stability in exchange for flexibility. For example, the GRU integrates inputs into the recurrent states in sequential order, has explicit forgetting mechanics, and keeps recurrent states bounded using saturating activation functions. These strong inductive biases curtail the model search space, improve training stability and sample efficiency, and provide a more "user-friendly" training experience. If inductive biases are indeed responsible for improved performance, can we better leverage them to improve memory in RL?

**Contributions** We introduce a memory model that summarizes a trajectory into a latent Markov state for a downstream policy. To enhance training stability and efficiency, we employ strong inductive priors inspired by computational psychology. Our model can replace RNNs in recurrent RL algorithms with a single line of code while training nearly two orders of magnitude faster than RNNs. Our experiments demonstrate that our model attains greater reward in both on-policy and off-policy settings and across various POMDP task suites, with similar sample efficiency to RNNs.

## 2 Related Work

RL and SL models differ in their computational requirements. While in SL the model training duration is primarily influenced by the forward and backward passes, RL produces training data through numerous inference workers interacting with the environment step by step. Consequently, it's imperative for RL memory to be both fast and efficient during both *training* and *inference*. However, for memory models, there is often an efficiency trade-off between between the training and inference stages, as well as a trade-off between time and space complexity.

**Recurrent Models** Recurrent models like LSTM [Hochreiter and Schmidhuber, 1997], GRUs [Chung et al., 2014], Legendre Memory Units (LMUs) [Voelker et al., 2019], and Independent RNNs [Li et al., 2018] are slow to train but fast at inference. Each recurrent state $\boldsymbol{S}$ over a sequence must be computed sequentially

$$\boldsymbol{y}_j, \boldsymbol{S}_j = f(\boldsymbol{x}_j, \boldsymbol{S}_{j-1}), \quad j \in [1, \ldots, n] \tag{1}$$

where $\boldsymbol{x}_j, \boldsymbol{y}_j$ are the inputs and outputs respectively at time $j$, and $f$ updates the state $\boldsymbol{S}$ incrementally. The best-case computational complexity of recurrent models scales linearly with the length of the sequence and such models *cannot be parallelized over the time dimension*, making them prohibitively slow to train over long sequences. On the other hand, such models are quick at inference, exhibiting constant-time complexity per inference timestep and constant memory usage over possibly infinite sequences.

**Parallel Models** Parallel or "batched" models like temporal CNNs [Bai et al., 2018] or transformers [Vaswani et al., 2017] do not rely on a recurrent state and can process an entire sequence in parallel i.e.,

$$\boldsymbol{y}_j = f(\boldsymbol{x}_1, \ldots, \boldsymbol{x}_j), \quad j \in [1, \ldots, n] \tag{2}$$

Given the nature of GPUs, these models exhibit faster training than recurrent models. Unfortunately, such models require storing all or a portion of the trajectory, preventing their use on long or infinite

sequences encountered during inference. Furthermore, certain parallel models like transformers require quadratic space and $n$ comparisons *at each timestep* during inference. This limits both the number of inference workers and their speed, resulting in inefficiencies, especially for on-policy algorithms.

**Hybrid Models** Linear transformers [Katharopoulos et al., 2020, Schlag et al., 2021, Su et al., 2021] or state space models [Gu et al., 2021, 2020] provide the best of both worlds by providing equivalent recurrent and closed-form (parallel) formulations

$$\boldsymbol{S}_j = f(\boldsymbol{x}_j, \boldsymbol{S}_{j-1}) = g(\boldsymbol{x}_1, \ldots, \boldsymbol{x}_j), \quad j \in [1, \ldots, n] \tag{3}$$

Training employs the batched formula to leverage GPU parallelism, while inference exploits recurrent models' low latency and small memory footprint. Thus, hybrid models are well-suited for RL because they provide both fast training and fast inference, which is critical given the poor sample efficiency of RL. However, recent findings show that common hybrid models typically underperform in comparison to RNNs on POMDPs [Morad et al., 2023].

## 3 Problem Statement

We are given a sequence of actions and observations $(\boldsymbol{o}_1, \boldsymbol{0}), (\boldsymbol{o}_2, \boldsymbol{a}_1), \ldots (\boldsymbol{o}_n, \boldsymbol{a}_{n-1})$ up to time $n$. Let the trajectory $\boldsymbol{X}$ be some corresponding encoding $\boldsymbol{x} = \varepsilon(\boldsymbol{o}, \boldsymbol{a})$ of each action-observation pair

$$\boldsymbol{X}_n = [\boldsymbol{x}_1, \ldots, \boldsymbol{x}_n] = [\varepsilon(\boldsymbol{o}_1, \boldsymbol{0}), \ldots, \varepsilon(\boldsymbol{o}_n, \boldsymbol{a}_{n-1})]. \tag{4}$$

Our goal is to summarize $\boldsymbol{X}_n$ into a latent Markov state $\boldsymbol{y}_n$ using some function $f$. For the sake of both space and time efficiency, we restrict our model to the space of *hybrid* memory models. Then, our task is to find an $f$, such that

$$\boldsymbol{y}_n, \boldsymbol{S}_n = f(\boldsymbol{X}_{k:n}, \boldsymbol{S}_{k-1}), \tag{5}$$

where $\boldsymbol{X}_{k:n}$ is shorthand for $\boldsymbol{x}_k, \boldsymbol{x}_{k+1}, \ldots, \boldsymbol{x}_n$. Note that by setting $k = n$, we achieve a one-step recurrent formulation that one would see in an RNN, i.e., $\boldsymbol{y}_n, \boldsymbol{S}_n = f(\boldsymbol{X}_{n:n}, \boldsymbol{S}_{n-1}) = f(\boldsymbol{x}_n, \boldsymbol{S}_{n-1})$.

## 4 Background

In the search for inductive priors to constrain our memory model search space, we turn to the field of computational psychology. In computational psychology, we use the term *trace* to refer to the physical embodiment of a memory – that is, the discernable alteration in the brain's structure before and after a memory's formation. Computational psychologists model long-term memory as a collection of traces, often in matrix form

$$\boldsymbol{S}_n = [\boldsymbol{x}_1, \ldots, \boldsymbol{x}_n], \tag{6}$$

where $\boldsymbol{x}_j$ are individual traces represented as column vectors and $\boldsymbol{S}_n$ is the memory at timestep $n$ [Kahana, 2020].

**Composite Memory** Composite memory [Galton, 1883] approximates memory as a lossy blending of individual traces $\boldsymbol{x}$ via summation, providing an explanation for how a lifetime of experiences can fit within a fixed-volume brain. Murdock [1982] expresses this blending via the recurrent formula

$$\boldsymbol{S}_n = \gamma \boldsymbol{S}_{n-1} + \boldsymbol{B}_n \boldsymbol{x}_n, \tag{7}$$

where $\gamma \in (0, 1)$ is the forgetting parameter and $\boldsymbol{B}_n$ is a diagonal matrix sampled from a Bernoulli distribution, determining a subset of $\boldsymbol{x}_n$ to add to memory. In Murdock's model, $\boldsymbol{S}$ is a vector, not a matrix. We can expand or "unroll" Murdock's recurrence relation, rewriting it in a closed form

$$\boldsymbol{S}_n = \gamma^n \boldsymbol{S}_0 + \gamma^{n-1} \boldsymbol{B}_1 \boldsymbol{x}_1 + \ldots \gamma^0 \boldsymbol{B}_n \boldsymbol{x}_n, \quad 0 < \gamma < 1, \tag{8}$$

making it clear that memory decays exponentially with time. Armed with equivalent recurrent and parallel formulations of composite memory, we can begin developing a hybrid memory model.

**Contextual Drift**  It is vital to note that incoming traces $\boldsymbol{x}$ capture raw sensory data and lack *context*. Context, be it spatial, situational, or temporal, differentiates between seemingly identical raw sensory inputs, and is critical to memory and decision making. The prevailing theory among computational psychologists is that contextual and sensory information are mixed via an outer product [Howard and Kahana, 2002]

$$\hat{\boldsymbol{x}}_n = \boldsymbol{x}_n \boldsymbol{\omega}_n^\top \tag{9}$$

$$\boldsymbol{\omega}_n = \rho \boldsymbol{\omega}_{n-1} + \boldsymbol{\eta}_n, \quad 0 < \rho < 1, \tag{10}$$

where $\boldsymbol{\eta}_n$ is the contextual state at time $n$, and $\rho$ ensures smooth or gradual changes between contexts. Although context spans many domains, we focus solely on temporal context.

## 5  Fast and Forgetful Memory

Fast and Forgetful Memory (FFM) is a hybrid memory model based on theories of composite memory and contextual drift. It is composed of two main components: a *cell* and an *aggregator*. The cell receives an input and recurrent state $\boldsymbol{x}, \boldsymbol{S}$ and produces a corresponding output and updated recurrent state $\boldsymbol{y}, \boldsymbol{S}$ (Figure 1). The aggregator resides within the cell and is responsible for updating $\boldsymbol{S}$ (Figure 2). We provide the complete equations for both the aggregator and cell and end with the reasoning behind their design.

**Aggregator**  The aggregator computes a summary $\boldsymbol{S}_n$ of $\boldsymbol{X}_{1:n}$, given a recurrent state $\boldsymbol{S}_{k-1}$ and inputs $\boldsymbol{X}_{k:n}$

$$\boldsymbol{S}_n = \boldsymbol{\gamma}^{n-k+1} \odot \boldsymbol{S}_{k-1} + \sum_{j=k}^{n} \boldsymbol{\gamma}^{n-j} \odot (\boldsymbol{x}_j \mathbf{1}_c^\top), \quad \boldsymbol{S}_n \in \mathbb{C}^{m \times c} \tag{11}$$

$$\boldsymbol{\gamma}^t = \left( \exp\left(-\boldsymbol{\alpha}\right) \exp\left(-i\boldsymbol{\omega}\right)^\top \right)^{\odot t} = \begin{bmatrix} \exp\left(-t(\alpha_1 + i\omega_1)\right) & \dots & \exp\left(-t(\alpha_1 + i\omega_c)\right) \\ \vdots & & \vdots \\ \exp\left(-t(\alpha_m + i\omega_1)\right) & \dots & \exp\left(-t(\alpha_m + i\omega_c)\right) \end{bmatrix} \in \mathbb{C}^{m \times c}. \tag{12}$$

where $\odot$ is the Hadamard product (or power), $m$ is the trace size, $c$ is the context size, and $\boldsymbol{\alpha} \in \mathbb{R}_+^m, \boldsymbol{\omega} \in \mathbb{R}^c$ are trainable parameters representing decay and context respectively. Multiplying column a vector by $\mathbf{1}_c^\top$ "broadcasts" or repeats the column vector $c$ times.

In Appendix C, we derive a memory-efficient and numerically stable closed form solution to compute all states $\boldsymbol{S}_{k:n}$ in parallel over the time dimension. Since our model operates over relative time, we map absolute time $k, \dots, n$ to relative time $p \in 0, \dots, t$, where $t = n - k$. The closed form to compute a state $\boldsymbol{S}_{k+p}$ given $\boldsymbol{S}_{k-1}$ is then

$$\boldsymbol{S}_{k+p} = \boldsymbol{\gamma}^{p+1} \odot \boldsymbol{S}_{k-1} + \boldsymbol{\gamma}^{p-t} \odot \sum_{j=0}^{p} \boldsymbol{\gamma}^{t-j} \odot (\boldsymbol{x}_{k+j} \mathbf{1}_c^\top), \quad \boldsymbol{S}_{k+p} \in \mathbb{C}^{m \times c}. \tag{13}$$

We can rewrite the aggregator's closed form (Equation 13) in matrix notation to highlight its time-parallel nature, computing all states $\boldsymbol{S}_{k:n} = \boldsymbol{S}_k, \boldsymbol{S}_{k+1}, \dots, \boldsymbol{S}_{k+n}$ at once

$$\boldsymbol{S}_{k:n} = \begin{bmatrix} \boldsymbol{\gamma}^1 \\ \vdots \\ \boldsymbol{\gamma}^{t+1} \end{bmatrix} \odot \begin{bmatrix} \boldsymbol{S}_{k-1} \\ \vdots \\ \boldsymbol{S}_{k-1} \end{bmatrix} + \begin{bmatrix} \boldsymbol{\gamma}^{-t} \\ \vdots \\ \boldsymbol{\gamma}^0 \end{bmatrix} \odot \begin{bmatrix} \left( \sum_{j=0}^{0} \boldsymbol{\gamma}^{t-j} \odot \left( \boldsymbol{x}_{k+j} \mathbf{1}_c^\top \right) \right) \\ \vdots \\ \left( \sum_{j=0}^{t} \boldsymbol{\gamma}^{t-j} \odot \left( \boldsymbol{x}_{k+j} \mathbf{1}_c^\top \right) \right) \end{bmatrix}, \quad \boldsymbol{S}_{k:n} \in \mathbb{C}^{(t+1) \times m \times c}. \tag{14}$$

The cumulative sum term in the rightmost matrix can be distributed across $t$ processors, each requiring $O(\log t)$ time using a prefix sum or scan [Harris et al., 2007].

**Cell**  The aggregator alone is insufficient to produce a Markov state $\boldsymbol{y}$. $\boldsymbol{S}$ is complex-valued and contains a large amount of information that need not be present in $\boldsymbol{y}$ at the current timestep, making

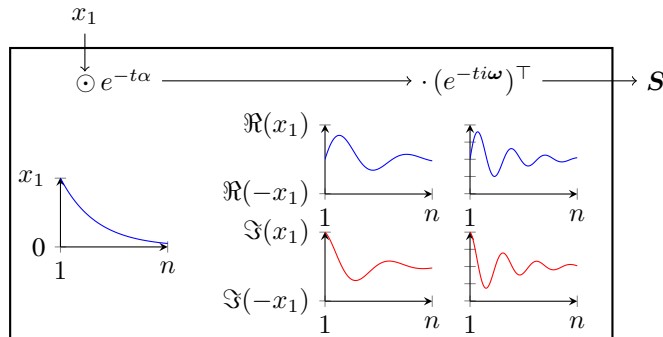

Figure 1: A detailed example of aggregator dynamics for $m = 1, c = 2$, showing how a one-dimensional input $x_1$ contributes to the recurrent state $S$ over time $t = [1, n]$. At the first timestep, $S_1$ is simply $x_1$. Over time, the contribution of $x_1$ to $S$ undergoes exponential decay via the $\alpha$ term (forgetting) and oscillations via the $\omega$ term (temporal context). In this example, the contribution from $x_1$ to $S_n$ approaches zero at time $n$.

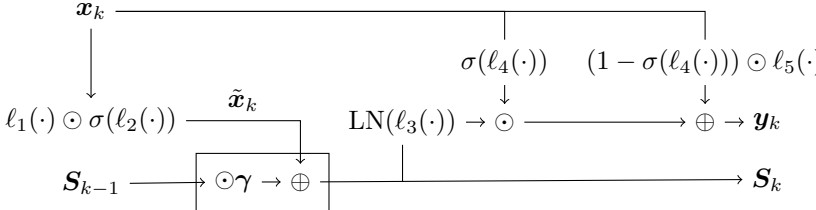

Figure 2: A visualization of an FFM cell running recurrently for a single timestep (i.e., inference mode). Inputs $x_k, S_{k-1}$ go through various linear layers ($\ell$), hadamard products ($\odot$), addition ($\oplus$), normalization (LN), and sigmoids ($\sigma$) to produce outputs $y_k, S_k$. The boxed region denotes the aggregator, which decays and shifts $S_{k-1}$ in time via $\gamma$. During training, the aggregator computes $S_k, \ldots, S_n$ in parallel so that we can compute all $y_k, \ldots, y_n$ in a single forward pass.

it cumbersome to interpret for a downstream policy. Furthermore, $X$ could benefit from additional preprocessing. The cell applies input gating to $X$ and extracts a real-valued Markov state $y$ from $S$.

$$\tilde{x}_{k:n} = \ell_1(x_{k:n}) \odot \sigma(\ell_2(x_{k:n})), \qquad\qquad \tilde{x}_{k:n} \in \mathbb{R}^{(t+1)\times m} \quad (15)$$

$$S_{k:n} = \text{Agg}(\tilde{x}_{k:n}, S_{k-1}), \qquad\qquad S_{k:n} \in \mathbb{C}^{(t+1)\times m \times c} \quad (16)$$

$$z_{k:n} = \ell_3(\text{Flatten}(\Re[S_{k:n}] \,||\, \Im[S_{k:n}])), \qquad\qquad z_{k:n} \in \mathbb{R}^{(t+1)\times d} \quad (17)$$

$$y_{k:n} = \text{LN}(z_{k:n}) \odot \sigma(\ell_4(x_{k:n})) + \ell_5(x_{k:n}) \odot (1 - \sigma(\ell_4(x_{k:n})). \quad y_{k:n} \in \mathbb{R}^{(t+1)\times d} \quad (18)$$

Agg represents the aggregator (Equation 14) and $\ell$ represents linear layers with mappings $\ell_1, \ell_2 : \mathbb{R}^d \to \mathbb{R}^m, \ell_3 : \mathbb{R}^{m \times 2c} \to \mathbb{R}^d, \ell_4, \ell_5 : \mathbb{R}^d \to \mathbb{R}^d$. $\Re, \Im$ extract the real and imaginary components of a complex number as reals, Flatten reshapes a matrix ($m \times c \to mc$) and $||$ is the concatenation operator. LN is nonparametric layer norm, and $\sigma$ is sigmoid activation. Equation 15 applies input gating, Equation 16 computes the recurrent states, Equation 17 projects the state into the real domain, and Equation 18 applies output gating.

**Modeling Inductive Biases**   To justify FFM's architecture, we detail its connections to computational psychology. Drawing from the theory of composite memory, we integrate sigmoidal gating to $x$ in Equation 15, approximating the sparsity-driven Bernoulli term, $B$, presented in (Equation 7). This inductive bias represents that only a small subset of each sensory experience is stored in memory.

Following the blending constraint imposed by composite memory (Equation 8), we sum the traces together (Equation 11). We suspect this blending prior yields a model that is more robust to the bad policy updates that plague RL, compared to RNNs that apply nonlinear and destructive transforms (e.g. deletion) to the recurrent state.

In Equation 12, the exponential component $\exp(-t\boldsymbol{\alpha})$ ensures that traces are forgotten according to composite memory decay from Equation 8. We choose an exponential decay $e^{-|\boldsymbol{\alpha}|}$, approximating Murre and Dros [2015]. This inductive bias asymptotically decays traces, reducing the size of the model optimization space. This is in direct contrast to recent SL memory models which aim to store as much information as possible for as long as possible [Gu et al., 2021, Voelker et al., 2019, Schlag et al., 2021]. In our ablations, we show that forgetting is critical in RL.

Traditionally, contextual coding and composite memory are separate mechanisms, but we incorporate both through $\boldsymbol{\gamma}$. The $\exp(-i\boldsymbol{\omega}t)$ component of $\boldsymbol{\gamma}$ in Equation 12 applies temporal context, enabling relative-time reasoning. Following contextual coding theory, we take the outer product of a trace vector, in our case decayed by $\boldsymbol{\alpha}$, with a context vector produced by $\boldsymbol{\omega}$. We can rewrite $\boldsymbol{\gamma}^t = \boldsymbol{\gamma}^{t-1} \odot \boldsymbol{\gamma}$, mirroring the gradual recurrent context changes from Equation 10.

The recurrent state $\boldsymbol{S}$ is a complex-valued matrix, so we project it back to the real domain as $\boldsymbol{z}$ (Equation 17). $\boldsymbol{z}$ can be large for dimensions where $\boldsymbol{\alpha}$ is near zero (minimal decay), so we find it crucial to apply layer normalization in Equation 18. Finally, we apply a gated residual connection to improve convergence speed, letting gradient descent find the optimal mixture of input $\boldsymbol{x}$ to memory $\boldsymbol{z}$ for the Markov state $\boldsymbol{y}$.

**The Mathematics Underpinning FFM**  The goal of FFM is to produce a memory mechanism which can be parallelized across the time dimension. As discussed in section 4, we can achieve this through a summation of sensory information mixed by some arbitrary function $\phi$ with a temporal context $\boldsymbol{\omega}_j = \psi(j)$, generating an aggregator of the form

$$\boldsymbol{S}_n = \sum_{j=0}^{n} \phi(\psi(j), \boldsymbol{x}_j). \tag{19}$$

Any implementation under this formulation would provide memory with temporal context for *absolute* time. However, in order to maximize generalization, we wish to operate over *relative* time. This introduces a problem, as the *relative* temporal context $\psi(n-j)$ associated with each $x_j$ must be updated at each timestep. Therefore, we must find a function $h$ that updates each temporal context with some offset: $h(\psi(j), \psi(k)) = \psi(j+k)$. Solving for $h$, we get: $h(j,k) = \psi(\psi^{-1}(j) + \psi^{-1}(k))$. If we also set $\phi = h$, then applying a temporal shift $h(\cdot, k)$ to each term in the sum yields the same result as initially inserting each $x_j$ with $\psi(j+k)$:

$$h(\phi(\psi(j), x_j), k) = \phi(\psi(j+k), x_j) \tag{20}$$

Consequently, it is possible to update the terms at each timestep to reflect the relative time $n-j$. However, it is intractable to recompute $n$ relative temporal contexts at each timestep, as that would result in $n^2$ time and space complexity for a sequence of length $n$. To apply temporal context in a batched manner in linear space, we can leverage the distributive property. If we select $\phi(a,b) = a \cdot b$, then the update distributes over the sum, updating all of the terms simultaneously. In other words, we can *update the context of $n$ terms in constant time and space* (see Appendix C for the full derivation). As $\psi$ is defined as a function of $\phi$, we can solve to find $\psi(t) = e^{\xi t}$ (although any exponential constitutes a solution, we select the natural base). This yields an aggregator of the form

$$\boldsymbol{S}_n = \sum_{j=0}^{n} e^{\xi(n-j)} \boldsymbol{x}_j. \tag{21}$$

If $\xi \in \mathbb{R}_+, \boldsymbol{x}_j \neq 0$, then the recurrent state will explode over long sequences: $\lim_{n \to \infty} e^{\xi(n-j)} \boldsymbol{x}_j = \infty$, so the real component of $\xi$ should be negative. Using different decay rates $\xi_j, \xi_k \in \mathbb{R}_-$, we can deduce the temporal ordering between terms $x_j$ and $x_k$. Unfortunately, this requires both $x_j, x_k$ eventually decay to zero – what if there are important memories we do not want to forget, while simultaneously retaining their ordering? If $\xi$ is imaginary, then we can determine the relative time between $\boldsymbol{x}_j, \boldsymbol{x}_k$ as the terms oscillate indefinitely without decaying. Thus, we use a complex $\xi$ with a negative real component, combining both forgetting and long-term temporal context. In the following paragraphs, we show that with a complex $\xi$, the FFM cell becomes a *universal approximator of convolution*.

**Universal Approximation of Convolution**  Here, we show that FFM can approximate any temporal convolutional. Let us look at the $\boldsymbol{z}$ term from Equation 17, given the input $\tilde{\boldsymbol{x}}$, with a slight change

| Model | Training | | Inference | |
|---|---|---|---|---|
| | Parallel Time | Space | Time | Space |
| RNN | $O(n)$ | $\boldsymbol{O(n)}$ | $\boldsymbol{O(1)}$ | $\boldsymbol{O(1)}$ |
| Transformer | $\boldsymbol{O(\log n)}$ | $O(n^2)$ | $O(n)$ | $O(n^2)$ |
| FFM (ours) | $\boldsymbol{O(\log n)}$ | $\boldsymbol{O(n)}$ | $\boldsymbol{O(1)}$ | $\boldsymbol{O(1)}$ |

Table 1: The time and space complexity of memory models for a sequence of length $n$ (training), or computing a single output recurrently during a rollout (inference).

in notation to avoid the overloaded subscript notation: $\tilde{\boldsymbol{x}}(\tau) = \tilde{\boldsymbol{x}}_\tau$, $\boldsymbol{z}(n) = \boldsymbol{z}_n$. For a sequence 1 to $n$, $\boldsymbol{z}(n)$, the precursor to the Markov state $\boldsymbol{y}_n$, can be written as:

$$\boldsymbol{z}(n) = \boldsymbol{b} + \boldsymbol{A} \sum_{\tau=1}^{n} \tilde{\boldsymbol{x}}(\tau) \odot \exp\left(-\tau\boldsymbol{\alpha}\right) \exp\left(-\tau i \boldsymbol{\omega}^\top\right). \tag{22}$$

where $\boldsymbol{A}, \boldsymbol{b}$ is the weight and bias from linear layer $\ell_3$, indexed by subscript. Looking at a single input dimension $k$ of $\tilde{\boldsymbol{x}}$, we have

$$\boldsymbol{z}(n) = \boldsymbol{b} + \sum_{j=kcm}^{(k+1)cm} \boldsymbol{A}_j \sum_{\tau=1}^{n} \tilde{x}_k(\tau) \exp\left(-\tau(i\boldsymbol{\omega} + \alpha_k)\right) \tag{23}$$

$$= \boldsymbol{b} + \sum_{\tau=1}^{n} \tilde{x}_k(\tau) \sum_{j=kcm}^{(k+1)cm} \boldsymbol{A}_j \exp\left(-\tau(i\boldsymbol{\omega} + \alpha_k)\right) \tag{24}$$

Equation 24 is a temporal convolution of $\tilde{x}_k(t)$ using a Fourier Series filter with $c$ terms ($\boldsymbol{\omega} \in \mathbb{C}^c$), with an additional learnable "filter extent" term $\alpha_k$. The Fourier Series is a universal function approximator, so FFM can approximate any convolutional filter over the signal $\tilde{\boldsymbol{x}}(\tau)$. Appendix A further shows how this is related Laplace transform. Unlike discrete convolution, we do not need to explicitly store prior inputs or engage in zero padding, resulting in better space efficiency. Furthermore, the filter extent $\alpha_k$ is learned and dynamic – it can expand for sequences with long-term temporal dependencies and shrink for tasks with short-term dependencies. Discrete temporal convolution from methods like Bai et al. [2018] use a fixed-size user-defined filter extent.

**Interpretability** Unlike other memory models, FFM is interpretable. Each dimension in $\boldsymbol{S}$ has a known decay rate and contextual period. We can determine trace durability (how long a trace lasts) $\boldsymbol{t}_\alpha$ and the maximum contextual period $\boldsymbol{t}_\omega$ of each dimension in $\boldsymbol{S}$ via

$$\boldsymbol{t}_\alpha = \frac{\log(\beta)}{\boldsymbol{\alpha} \mathbf{1}_c^\top}, \ \boldsymbol{t}_\omega = \frac{2\pi}{\mathbf{1}_m \boldsymbol{\omega}} \tag{25}$$

where $\beta$ determines the strength at which a trace is considered forgotten. For example, $\beta = 0.01$ would correspond to the time when the trace $\boldsymbol{x}_k$ contributes 1% of its original value to $\boldsymbol{S}_n$. FFM can measure the relative time between inputs up to a modulo of $\boldsymbol{t}_\omega$.

## 6 Experiments and Discussion

We evaluate FFM on the two largest POMDP benchmarks currently available: POPGym [Morad et al., 2023] and the POMDP tasks from [Ni et al., 2022], which we henceforth refer to as POMDP-Baselines. For POPGym, we train a shared-memory actor-critic model using recurrent using Proximal Policy Optimization [Schulman et al., 2017]. For POMDP-Baselines, we train separate memory models for the actor and critic using recurrent Soft Actor Critic (SAC) [Haarnoja et al., 2018] and recurrent Twin Delayed DDPG (TD3) [Fujimoto et al., 2018]. We replicate the experiments from the POPGym and POMDP-Baselines papers as-is *without changing any hyperparameters*. We use a single FFM configuration across all experiments, except for varying the hidden and recurrent sizes to match the RNNs, and initialize $\boldsymbol{\alpha}, \boldsymbol{\omega}$ based on the max sequence length. See Appendix D for further details.

Table 2 lists memory baselines, Table 3 ablates each component of FFM, Figure 3 contains a summary of training statistics for all models (wall-clock train time, mean reward, etc), and Figure 4

| Name | Type | Paper/Description | Name | Type | Paper/Description |
|------|------|-------------------|------|------|-------------------|
| MLP | Basic | 2 layer perceptron | LMU | RNN | Voelker et al. [2019] |
| PosMLP | Basic | MLP w/ positional encoding | IndRNN | RNN | Li et al. [2018] |
| Stack | Basic | Mnih et al. [2015] | Elman | RNN | Elman [1990] |
| TCN | Conv | Bai et al. [2018] | GRU | RNN | Chung et al. [2014] |
| FWP | XFormer | Schlag et al. [2021] | LSTM | RNN | Hochreiter and Schmid- |
| FART | XFormer | Katharopoulos et al. [2020] | | | huber [1997] |
| S4D | Hybrid | Gu et al. [2021] | DNC | MANN | Graves et al. [2016] |

Table 2: The memory models used in the POPGym experiments, see Morad et al. [2023] for further details. Conv stands for convolution, XFormer stands for transformer, and MANN stands for memory augmented neural network.

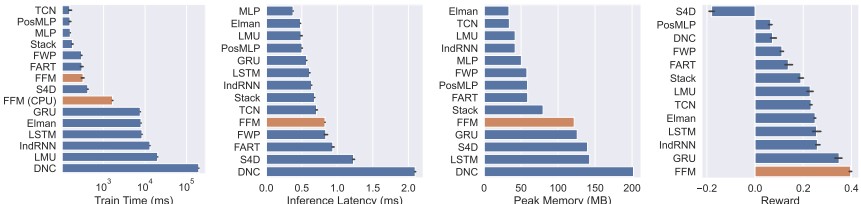

Figure 3: Training statistics computed over ten trials and one epoch of PPO, with FFM in orange. FFM trains nearly two orders of magnitude faster on the GPU than the fastest RNNs (note the log-scale x axis). Even on the CPU, FFM trains roughly one order of magnitude faster than RNNs on the GPU. FFM has sub-millisecond CPU inference latency, making it useful for on-policy algorithms. FFM memory usage is on-par with RNNs, providing scalability to long sequences. Despite efficiency improvements, FFM still attains greater episodic reward than other models on POPGym.

contains a detailed comparison of FFM against the best performing RNN and transformer on POPGym. Figure 5 evaluates FFM on the POMDP-Baselines tasks and Figure 6 investigates FFM interpretability. See Appendix E and Appendix F for more granular plots. Error bars in all cases denote the 95% bootstrapped confidence interval.

**Practicality and Robustness:** FFM demonstrates exceptional performance on the POPGym and POMDP-Baselines benchmarks, achieving the highest mean reward as depicted in Figure 3 and Figure 5 *without any changes to the default hyperparameters*. FFM also executes a PPO training epoch nearly two orders of magnitude faster than a GRU (Figure 3, Table 1). Without forgetting, FFM underperforms the GRU, showing that forgetting is the most important inductive prior (Table 3). We use a single FFM configuration for each benchmark, demonstrating it is reasonably insensitive to hyperparameters. All in all, FFM outperforms all other models on average, across three RL algorithms and 52 tasks. Surprisingly, there are few occurrences where FFM is noticeably worse than others, suggesting it is robust and a good general-purpose model.

**Explainability and Prior Knowledge:** Figure 6 demonstrates that FFM learns suitable and interpretable decay rates and context lengths. We observe separate memory modes for the actor and

|   | FFM | FFM-NI | FFM-NO | FFM-NC | FFM-FC | FFM-ND | FFM-FD | FFM-HP |
|---|-----|--------|--------|--------|--------|--------|--------|--------|
| $\mu$ | **0.399** | 0.392 | 0.395 | 0.382 | 0.383 | 0.333 | 0.390 | 0.395 |
| $\sigma$ | **0.005** | 0.001 | 0.006 | 0.005 | 0.005 | 0.005 | 0.006 | 0.010 |

Table 3: Ablating FFM components over all POPGym tasks. FFM-NI removes input gating, FFM-NO removes the output gating, FFM-NC does not use temporal context, FFM-FC uses fixed (non-learned) context, FFM-ND does not use decay, and FFM-FD uses fixed (non-learned) decay. FFM-HP uses the Hadamard product instead of the outer product to compute $\gamma$ in Equation 12. FFM-HP requires additional parameters but ensures independence between rows and columns of the state. FFM-ND (no decay) validates our assumption that forgetting is an important inductive bias.

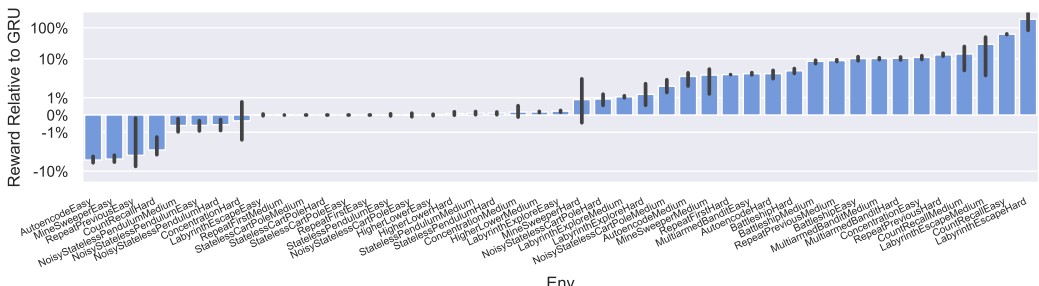

Figure 4: FFM compared to the GRU, the best performing model on POPGym. See Appendix F for comparisons across all other POPGym models, such as linear transformers. Error bars denote the bootstrapped 95% confidence interval over five trials. FFM noticeably outperforms state of the art on seventeen tasks, while only doing noticeably worse on four tasks.

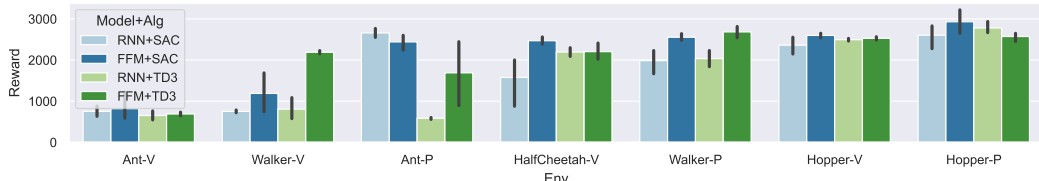

Figure 5: Comparing FFM with tuned RNNs on the continuous control tasks from POMDP-Baselines. Ni et al. [2022] selects the best performing RNN (either GRU or LSTM) and tunes hyperparameters on a per-task basis. Error bars denote the bootstrapped 95% confidence interval over five random seeds. The task suffix V or P denotes whether the observation is velocity-only (masked position) or position-only (masked velocity) respectively. Untuned FFM outperforms tuned RNNs on all but one task for SAC. For TD3, FFM meets or exceeds RNN performance on all but one task.

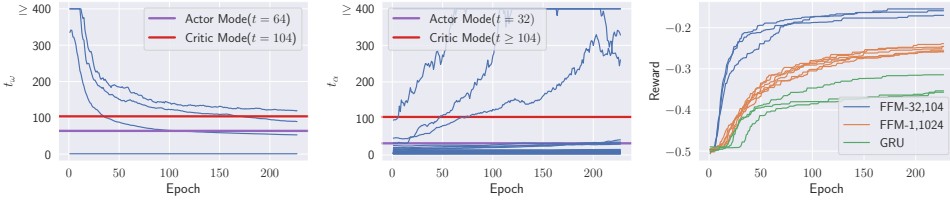

Figure 6: A case study on FFM explainability. We investigate the task RepeatPreviousMedium, where the episode length is 104 timesteps and the agent must output the observation from 32 timesteps ago. **(Left two)** We visualize $\gamma$ using trace durability $t_\alpha, \beta = 0.1$ and context period $t_\omega$. Both $t_\alpha, t_\omega$ demonstrate modes for the actor and critic, which we highlight in the figures. The critic requires the episode's current timestep to accurately predict the discounted return, while the actor just requires 32 timesteps of memory. **(Left)** The $t_\omega$ critic mode converges to the maximum episode length of 104. The actor mode converges to 2x the necessary period (perhaps because $\cos(\frac{2\pi}{t_\omega})$, the real component of $\exp -ti\omega$, is monotonic on the half-period $0.5 t_\omega$). **(Middle)** The $t_\alpha$ terms converge to 32 for the actor and a large value for the critic. **(Right)** We initialize $\alpha, \omega$ to encapsulate ideal actor and critic modes, such that $t_\alpha \in [32, 104], t_\omega \in [32, 104]$. We find FFM with informed initialization outperforms FFM with the original initialization of $t_\alpha \in [1, 1024], t_\omega \in [1, 1024]$.

critic, and we can initialize $\alpha, \omega$ using prior knowledge to improve returns (Figure 6). This sort of prior knowledge injection is not possible in RNNs, and could be useful for determining the number of "burn-in" steps for methods that break episodes into fixed-length sequences, like R2D2 [Kapturowski et al., 2019] or MEME [Kapturowski et al., 2023].

**Limitations and Future Work**

**Scaling Up:** We found that large context sizes $c$ resulted in decreased performance. We hypothesize that a large number of periodic functions results in a difficult to optimize loss landscape with many local extrema. Interestingly, transformers also employ sinusoidal encodings, which might explain the difficulty of training them in model-free RL. We tried multiple configurations of FFM cells in series, which helped in some tasks but often learned much more slowly. In theory, serial FFM cells could learn a temporal feature hierarchy similar to the spatial hierarchy in image CNNs. In many cases, serial FFM models did not appear fully converged, so it is possible training for longer could solve this issue.

**Additional Experiments:** Hyperparameter tuning would almost guarantee better performance, but would also result in a biased comparison to other models. We evaluated FFM with on-policy and off-policy algorithms but did not experiment with offline or model-based RL algorithms. In theory, FFM can run in continuous time or irregularly-spaced intervals simply by letting $t$ be continuously or irregular, but in RL we often work with discrete timesteps at regular intervals so we due not pursue this further.

**Numerical Precision:** FFM experiences a loss of numerical precision caused by the repeated multiplication of exponentials, resulting in very large or small numbers. Care must be taken to prevent overflow, such as upper-bounding $\alpha$. Breaking a sequence into multiple forward passes while propagating recurrent state (setting $n - k$ less than the sequence length) fixes this issue, but also reduces the training time efficiency benefits of FFM. We found FFM performed poorly using single precision floats, and recommend using double precision floats during multiplication by $\gamma$. We tested a maximum sequence length of 1024 per forward pass, although we could go higher by decreasing the maximum decay rate. With quadruple precision floats, we could process sequences of roughly 350,000 timesteps in a single forward pass with the max decay rate we used. Unfortunately, Pytorch does not currently support quads.

# 7   Conclusion

The inductive priors underpinning FFM are key to its success, constraining the optimization space and providing parallelism-enabling structure. Unlike many memory models, FFM is interpretable and even provides tuning opportunities based on prior knowledge about individual tasks. FFM provides a low-engineering cost "plug and play" upgrade to existing recurrent RL algorithms, improving model efficiency and reward in partially observable model-free RL with a one-line code change.

# 8   Acknowledgements

Steven Morad and Stephan Liwicki gratefully acknowledge the support of Toshiba Europe Ltd. Ryan Kortvelesy and Amanda Prorok were supported in part by ARL DCIST CRA W911NF-17-2-0181.

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

## A  Relationship to the Laplace Transform

We often think of RL in terms of discrete timesteps, but the sum in Equation 11 could be replaced with an integral for continuous-time domains, where $\tilde{x}$ is a function of time:

$$\boldsymbol{S}(n) = \int_0^n \tilde{\boldsymbol{x}}(\tau) \odot \exp\left(-\tau\boldsymbol{\alpha}\right) \exp\left(-\tau i\boldsymbol{\omega}\right)^\top d\tau. \tag{26}$$

If we look at just a single dimension of $\tilde{\boldsymbol{x}}$, we have

$$\boldsymbol{S}(n) = \int_0^n \tilde{x}(\tau) \exp\left(-\tau(\alpha + i\boldsymbol{\omega})\right) d\tau, \tag{27}$$

which is equivalent to the Laplace Transform $L$ of a function $\tilde{x}(\tau)$. One way to interpret the connection between our aggregator and the Laplace transform is that the aggregator transforms the memory task into a pole-finding task in the S-plane, a well-known problem in control theory. When we train FFM for an RL task, we are attempting to find the poles and zeros that minimize the actor/critic loss for a batch of sequences.

## B  Weight Initialization

We find that our model is sensitive to initialization of the $\boldsymbol{\alpha}, \boldsymbol{\omega}$ values. We compute the upper limit for $\boldsymbol{\alpha}$ such that memory will retain $\beta = 0.01 = 1\%$ of its value after some elapsed number of timesteps we call $t_e$. This can be computed mathematically via

$$\frac{\log \beta}{t_e}. \tag{28}$$

The lower limit is set based on the maximum value a double precision float can represent, minus a small $\epsilon$

$$\frac{\log 1.79 \times 10^{308}}{t_e} - \epsilon. \tag{29}$$

Note that if needed, we can choose a smaller lower bound, but the input must be chunked into sequences of length $t_e$ and run in minibatches. Since we can compute minibatches recurrently, the gradient spans across all minibatches rather than being truncated like a quadratic transformer. Nonetheless, we did not need to do this in any of our experiments. We set $\boldsymbol{\alpha}$ to be linearly spaced between the computed limits

$$\boldsymbol{\alpha} = [\alpha_1, \ldots \alpha_j \ldots \alpha_n]^\top \tag{30}$$

$$\alpha_j = \frac{j}{m}\frac{\log \beta}{t_e} + (1 - \frac{j}{m})\left(\frac{\log 1.79 \times 10^{308}}{t_e} - \epsilon\right) \tag{31}$$

We initialize the $\boldsymbol{\omega}$ terms using a similar approach, with linearly spaced denominators between one and $t_e$

$$\boldsymbol{\omega} = 2\pi/[\omega_1, \omega_2, \ldots, \omega_n]^\top \tag{32}$$

$$\omega_j = \frac{j}{c} + (1 - \frac{j}{c})t_e \tag{33}$$

All other parameters are initialized using the default Pytorch initialization.

## C  Computing States in Parallel with Linear Space Complexity

Here, we show how the recurrent formulation can be computed in parallel. A naive implementation would use $O(n^2)$ space, but using a trick, we can accomplish this in $O(n)$ space.

Assume we have already computed hidden state $\boldsymbol{S}_{k-1}$ for some sequence. Our job is now to compute the next $n-k+1$ recurrent states $\boldsymbol{S}_k, \boldsymbol{S}_{k+1}, \ldots, \boldsymbol{S}_n$ in parallel. We can rewrite Equation 11 in matrix

form:

$$
\boldsymbol{S}_{k:n} = \begin{bmatrix} \boldsymbol{S}_k \\ \boldsymbol{S}_{k+1} \\ \boldsymbol{S}_{k+2} \\ \vdots \\ \boldsymbol{S}_n \end{bmatrix} = \begin{bmatrix} \boldsymbol{\gamma}^0 \odot (\boldsymbol{x}_k \mathbf{1}_c^\top) + \boldsymbol{\gamma}^1 \odot \boldsymbol{S}_{k-1} \\ \boldsymbol{\gamma}^0 \odot (\boldsymbol{x}_{k+1} \mathbf{1}_c^\top) + \boldsymbol{\gamma}^1 \odot (\boldsymbol{x}_k \mathbf{1}_c^\top) + \boldsymbol{\gamma}^2 \odot \boldsymbol{S}_{k-1} \\ \boldsymbol{\gamma}^0 \odot (\boldsymbol{x}_{k+2} \mathbf{1}_c^\top) + \boldsymbol{\gamma}^1 \odot (\boldsymbol{x}_{k+1} \mathbf{1}_c^\top) + \boldsymbol{\gamma}^2 \odot (\boldsymbol{x}_k \mathbf{1}_c^\top) + \boldsymbol{\gamma}^3 \odot \boldsymbol{S}_{k-1} \\ \vdots \\ \left( \sum_{j=k}^n \boldsymbol{\gamma}^j \odot (\boldsymbol{x}_{k+j} \mathbf{1}_c^\top) \right) + \boldsymbol{\gamma}^{n+1} \odot \boldsymbol{S}_{k-1} \end{bmatrix} \quad (34)
$$

We can write the closed form for the $p$th row of the matrix, where $0 \geq p \geq n - k$

$$
\boldsymbol{S}_{k+p} = \left( \sum_{j=0}^p \boldsymbol{\gamma}^{p-j} \odot (\boldsymbol{x}_{k+j} \mathbf{1}_c^\top) \right) + \boldsymbol{\gamma}^{p+1} \odot \boldsymbol{S}_{k-1} \quad (35)
$$

Unfortunately, it appears we will need to materialize $\frac{(n-k+1)^2}{2}$ terms:

$$
\boldsymbol{\gamma}^0 \odot (\boldsymbol{x}_k \mathbf{1}_c^\top) \quad (36)
$$
$$
\boldsymbol{\gamma}^1 \odot (\boldsymbol{x}_k \mathbf{1}_c^\top), \boldsymbol{\gamma}^0 \odot (\boldsymbol{x}_{k+1} \mathbf{1}_c^\top) \quad (37)
$$
$$
\vdots \quad (38)
$$
$$
\boldsymbol{\gamma}^{n-k} \odot (\boldsymbol{x}_k \mathbf{1}_c^\top), \boldsymbol{\gamma}^{n-k+1} \odot (\boldsymbol{x}_{k+1} \mathbf{1}_c^\top), \ldots \boldsymbol{\gamma}^0 \odot (\boldsymbol{x}_n \mathbf{1}_c^\top) \quad (39)
$$

However, we can factor out $\boldsymbol{\gamma}^{-p}$ from Equation 35 via a combination of the distributive property and the product of exponentials

$$
\boldsymbol{S}_{k+p} = \gamma^p \odot \left( \sum_{j=0}^p \boldsymbol{\gamma}^{-j} \odot (\boldsymbol{x}_{k+j} \mathbf{1}_c^\top) \right) + \boldsymbol{\gamma}^{p+1} \odot \boldsymbol{S}_{k-1} \quad (40)
$$

Now, each $\boldsymbol{\gamma}^j$ is associated with a single $\boldsymbol{x}_j$, requiring just $n - k + 1$ terms:

$$
\boldsymbol{\gamma}^0 \odot (\boldsymbol{x}_k \mathbf{1}_c^\top) \quad (41)
$$
$$
\boldsymbol{\gamma}^0 \odot (\boldsymbol{x}_k \mathbf{1}_c^\top), \boldsymbol{\gamma}^1 \odot (\boldsymbol{x}_{k+1} \mathbf{1}_c^\top) \quad (42)
$$
$$
\vdots \quad (43)
$$
$$
\boldsymbol{\gamma}^0 \odot (\boldsymbol{x}_k \mathbf{1}_c^\top), \boldsymbol{\gamma}^1 \odot (\boldsymbol{x}_{k+1} \mathbf{1}_c^\top), \ldots, \boldsymbol{\gamma}^n \odot (\boldsymbol{x}_n \mathbf{1}_c^\top) \quad (44)
$$

We can represent each of these rows as a slice of a single $n - k + 1$ length tensor, for a space complexity of $O(n - k)$ or for $k = 1$, $O(n)$.

Finally, we want to swap the exponent signs because it provides better precision when working with floating point numbers. Computing small values, then big values is more numerically stable than the other way around. We want to compute the inner sum using small numbers ($\boldsymbol{\gamma}^+$ results in small numbers while $\boldsymbol{\gamma}^-$ produces big numbers; $n - k$ is positive and $k - n$ is negative). We can factor $\boldsymbol{\gamma}^{k-n}$ out of the sum and rewrite Equation 40 as

$$
\boldsymbol{S}_{k+p} = \gamma^{k-n+p} \odot \left( \sum_{j=0}^p \boldsymbol{\gamma}^{n-k-j} \odot (\boldsymbol{x}_{k+j} \mathbf{1}_c^\top) \right) + \boldsymbol{\gamma}^{p+1} \odot \boldsymbol{S}_{k-1} \quad (45)
$$

If we let $t = n - k$, this is equivalent to Equation 13:

$$
\boldsymbol{S}_{k+p} = \boldsymbol{\gamma}^{p+1} \odot \boldsymbol{S}_{k-1} + \boldsymbol{\gamma}^{p-t} \odot \sum_{j=0}^p \boldsymbol{\gamma}^{t-j} \odot (\boldsymbol{x}_{k+j} \mathbf{1}_c^\top), \quad \boldsymbol{S}_{k+p} \in \mathbb{C}^{m \times c}. \quad (46)
$$

We also need to compute $2(n - k + 1)$ gamma terms: $\boldsymbol{\gamma}^{k-n}, \ldots \boldsymbol{\gamma}^{n-k+1}$, resulting in linear space complexity $O(n)$ for a sequence of length $n$.

# D Benchmark Details

We utilize the episodic reward and normalized episodic reward metrics. We record episodic rewards as a single mean for each epoch, and report the maximum reward over a trial.

We do not modify the hyperparameters for PPO, SAC, or TD3 from the original benchmark papers. We direct the reader to [Morad et al., 2023] for PPO hyperparameters and [Ni et al., 2022] for SAC and TD3 hyperparameters.

## D.1 Hardware and Efficiency Information

We trained on a server with a Xeon CPU running Torch version 1.13 with CUDA version 11.7, with consistent access to two 2080Ti GPUs. Wandb reports that we used roughly 161 GPU days of compute to produce the POMDP-Baselines results, 10 GPU days for the POPGym results, and 45 GPU days for the FFM ablation using POPGym. This results in a total of 216 GPU days. Since we ran four jobs per GPU, this corresponds to 54 days of wall-clock time for all experiments.

## D.2 PPO and POPGym Baselines

Morad et al. [2023] compares models along the recurrent state size, with a recurrent state size of 256. For fairness, We let $m = 32$ and $c = 4$, which results in a complex recurrent state of 128, which can be represented as a 256 dimensional real vector. We initialize $\boldsymbol{\alpha}, \boldsymbol{\omega}$ following Appendix B for $t_e = 1024, \beta = 0.01$. We run version 0.0.2 of POPGym, and compare FFM numerically with the MMER score from the paper in Table 4.

## D.3 SAC, TD3, and POMDP-Baselines

[Ni et al., 2022] does not compare along the recurrent state size, but rather the hidden size, while utilizing various hidden sizes $h$. In other words, the LSTM recurrent size is twice that of the GRU. We let $c = h/32$ and $m = h/c$, so that $mc = h$. This produces an equivalent FFM configuration to the POPGym baseline when $h = 128$, with equivalent recurrent size in bytes to the LSTM (or equivalent in dimensions to the GRU). The paper truncates episodes into segments of length 32 or 64 depending on the task, so we let $t_e = 128$ to ensure that information can persist between segments. Thanks to the determinism provided in the paper, readers should be able to reproduce our exact results using the random seeds $0, 1, 2, 3, 4$. We utilize separate memory modules for the actor and critic, as done in the paper. We base our experiments off of the most recent commit at the time of writing, 4d9cbf1.

| Model | MMER |
|---|---|
| MLP | 0.067 |
| PosMLP | 0.064 |
| FWP | 0.112 |
| FART | 0.138 |
| S4D | -0.180 |
| TCN | 0.233 |
| Fr.Stack | 0.190 |
| LMU | 0.229 |
| IndRNN | 0.259 |
| Elman | 0.249 |
| GRU | 0.349 |
| LSTM | 0.255 |
| DNC | 0.065 |
| **FFM** | **0.400** |

Table 4: MMER score comparison from the POPGym paper

# E   Benchmark Lineplots

We provide cumulative max reward lineplots in Figure 9, Figure 7, and Figure 8 for all the experiments we ran.

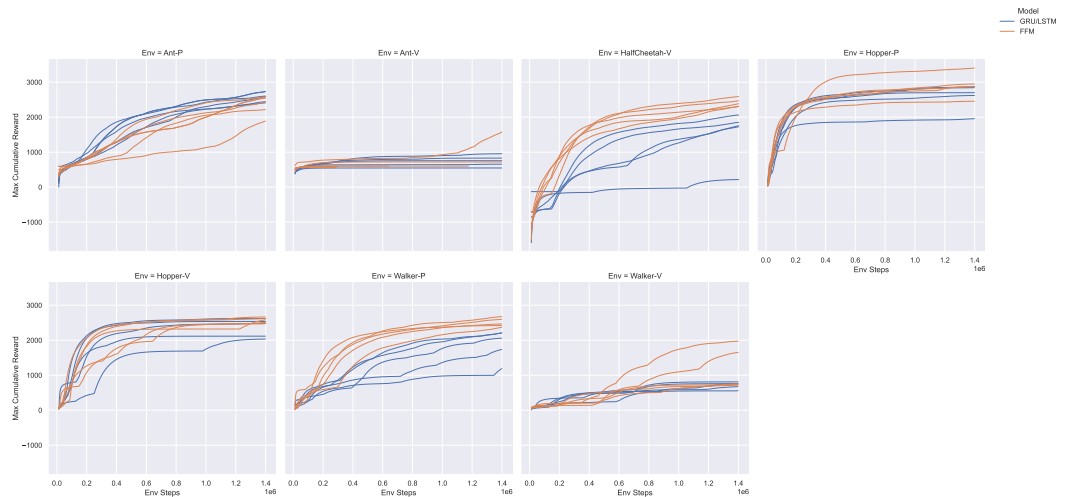

Figure 7: SAC lineplots for POMDP-Baselines, where each trial is plotted separately.

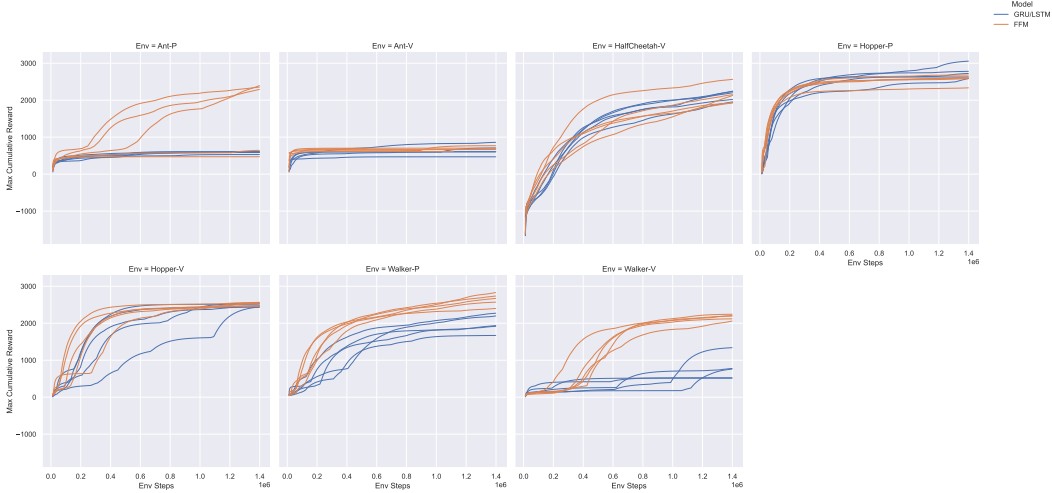

Figure 8: TD3 lineplots for POMDP-Baselines, where each trial is plotted separately.

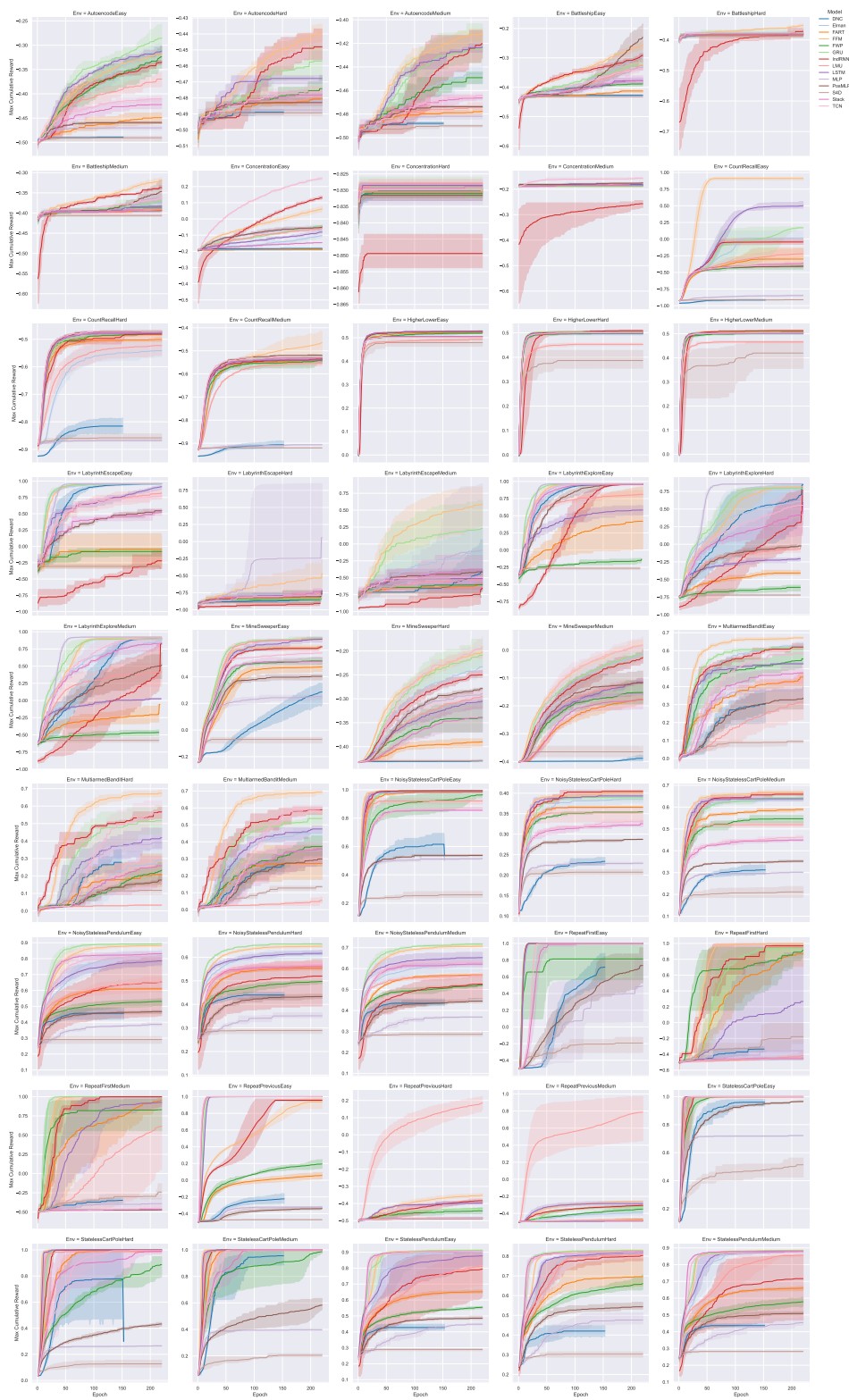

Figure 9: Lineplots for POPGym, where the shaded region represents the 95% bootstrapped confidence interval.

# F POPGym Comparisons by Model

We provide Figure 10, Figure 11, Figure 12 showing the relative FFM return compared to the other 12 POPGym models, including temporal convolution, linear transformers, and more.

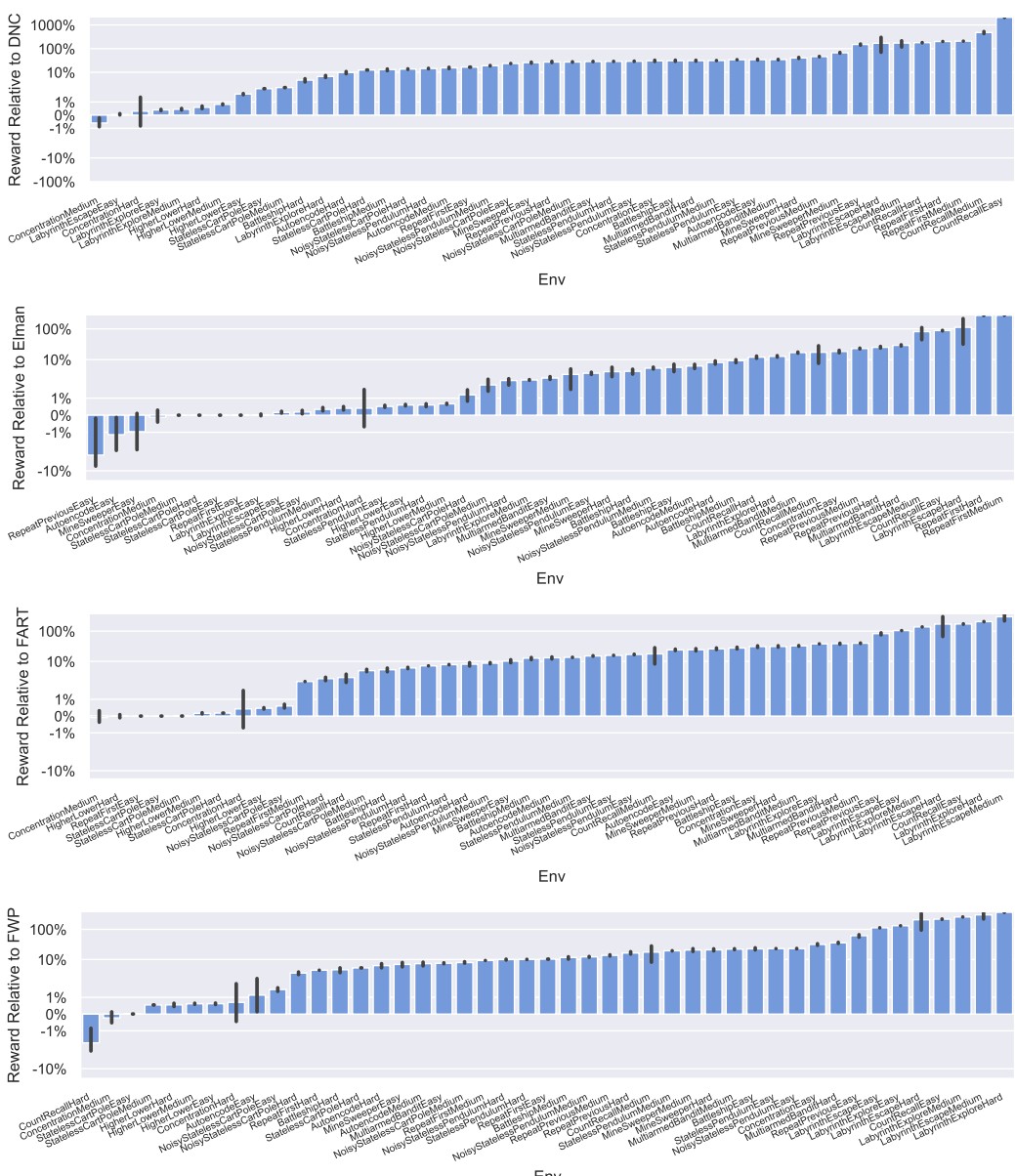

Figure 10: Relative POPGym returns compared to FFM.

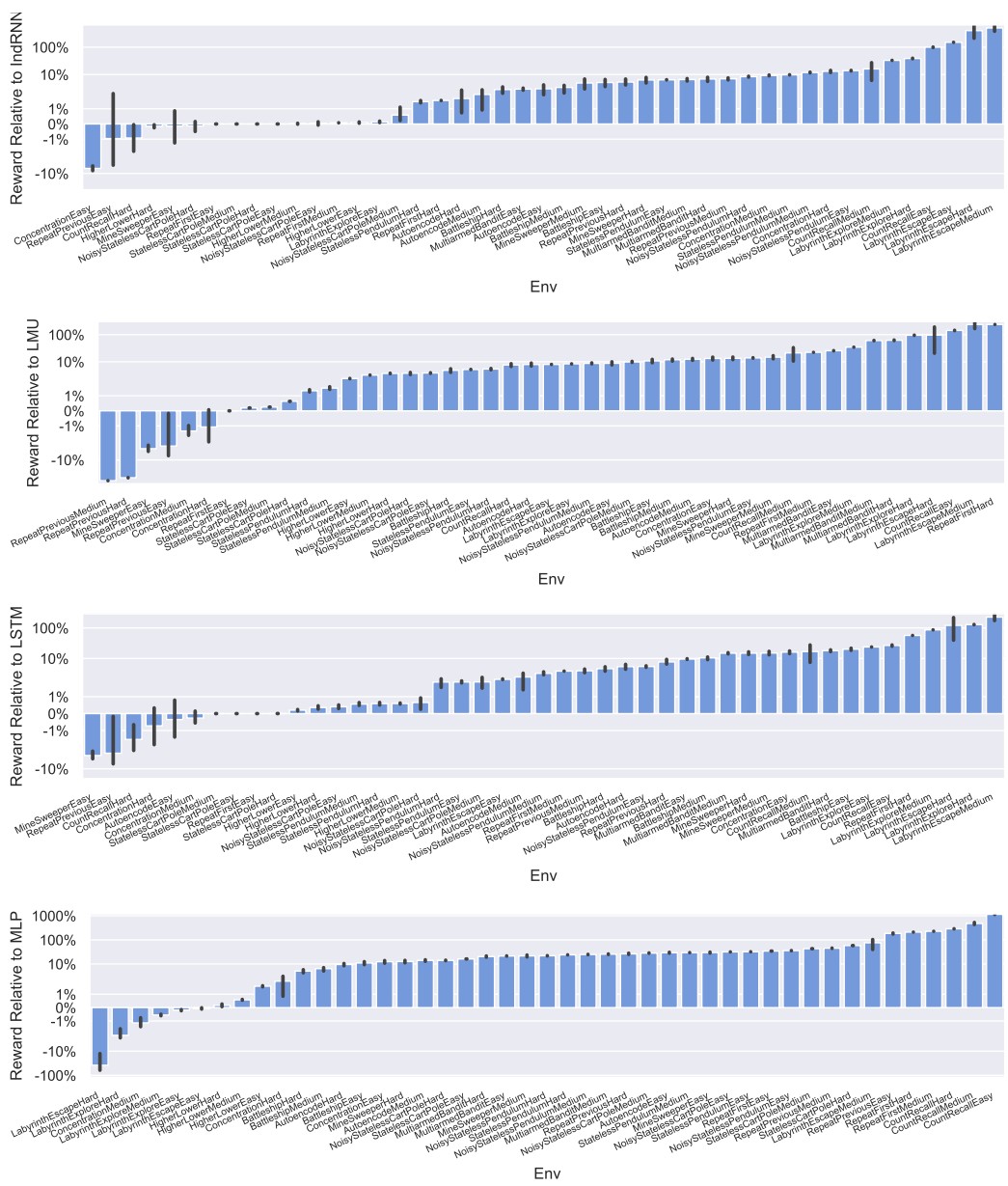

Figure 11: Relative POPGym returns compared to FFM.

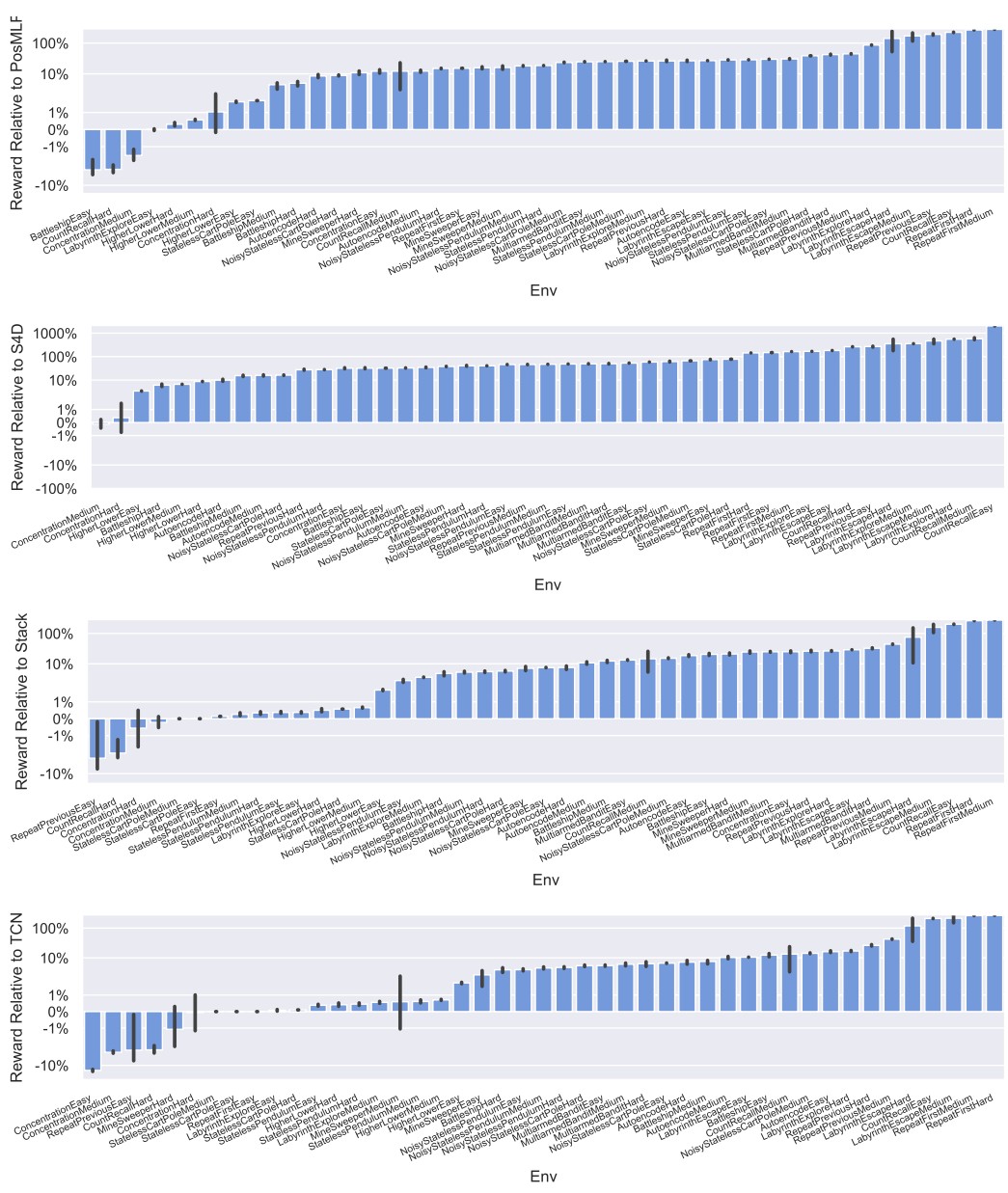

Figure 12: Relative POPGym returns compared to FFM.

# G   Efficiency Statistic Details

This section explains how we computed the bar plots showing efficiency statistics. We construct the POPGym models and perform one PPO training epoch. The train time metric corresponds to the time spent doing forward and backward passes over the data. One epoch corresponds to the epoch defined in POPGym: 30 minibatches of 65,336 transitions each, with an episode length of 1024, a hidden dimension of 128, and a recurrent dimension of 256. We do this 10 times and compute the mean and confidence interval. Torch GRU, LSTM, and Elman networks have specialized CUDA kernels, making them artificially faster than LMU and IndRNN which are written in Torch and require the use of for loops. We utilize the pure python implementation of these models, wrapping them in a for loop instead of utilizing their hand-designed CUDA kernels. We consider this a fair comparison since this makes the GRU, Elman, and LSTM networks still run slightly faster than IndRNN and LMU (Torch-native RNNs). FFM is also written in Torch and does not have access to specialized CUDA kernels. CUDA programmers more skilled than us could implement a custom FFM kernel that would see a speedup similar to the GRU/LSTM/Elman kernels.

To compute inference latency, we turn off Autograd and run inference for 1024 timesteps, computing the time for each forward pass. We do this 10 times and compute the mean and 95% confidence interval.

To compute memory usage, we utilize Torch's GPU memory tools and record the maximum memory usage at any point during the training statistic. Memory usage is constant between trials.

For reward, we take the mean reward over all environments, split by model and trial. We then report the mean and 95% confidence interval over trials and models.

