# OpenReview forum: "Reinforcement Learning with Fast and Forgetful Memory"
_NeurIPS.cc/2023/Conference — NeurIPS 2023 poster_

### Official Review · Reviewer_AuBN · 2023-06-19

**Soundness:** 2 fair
**Presentation:** 2 fair
**Contribution:** 2 fair
**Rating:** 4
**Confidence:** 4

**Summary:**

The paper proposes a new memory model for reinforcement learning agents, targeting POMDP setting. The memory (FFM) aggerates past data via an exponentially weighted summation operation where the weights are learnable parameters. The memory is used to compute the Markov state through sophisticated gating mechanisms. The paper evaluates FFM on 2 POMDP benchmarks and shows improvement over baselines such as GRU.

**Strengths:**

- The memory design is motivated by computational psychology
- The proposed memory is light-weight and fast

**Weaknesses:**

- The novelty is questionable. The main contribution is Eq. 11, and it is very similar to Eq. 1 in [1]. The difference is that the decay term here is learnable and consists of real and imaginary components. Table 2 reveals that non-learned decay also shows decent performance, which questions the need of learning the decay.
- The literature review lacks a line of research using MANN on RL [2,3,4]. They are memory models specifically designed for RL. It is fairer to compare FFM with them rather than comparing with general memory models such as DNC or GRU. Please note that general memory models may require heavy hyperparameter tuning and adjustment to make them work on RL, thus comparing with well-established memory-based RL methods is better. For example, Transformer needs some tricks to work on RL [5]. Please consider using this baseline instead of general Transformer models.
- The experimental results are not convincing. In POPGym, the gap between FFM and GRU is not clear. The ablation study in Table 2 indicates that the full model does not outperform other configurations significantly, especially when the std is high and only 3 runs are executed. It would suggest that the complicated design might not add much to the performance gain. In Fig. 5, there are many tasks FFM shows similar or worse results than RNN. It questions the benefit of using FFM.

[1] Ba, Jimmy, Geoffrey E. Hinton, Volodymyr Mnih, Joel Z. Leibo, and Catalin Ionescu. "Using fast weights to attend to the recent past." Advances in neural information processing systems 29 (2016).

[2] Wayne, Greg, Chia-Chun Hung, David Amos, Mehdi Mirza, Arun Ahuja, Agnieszka Grabska-Barwinska, Jack Rae et al. "Unsupervised predictive memory in a goal-directed agent." arXiv preprint arXiv:1803.10760 (2018).

[3]  Fortunato, Meire, Melissa Tan, Ryan Faulkner, Steven Hansen, Adrià Puigdomènech Badia, Gavin Buttimore, Charles Deck, Joel Z. Leibo, and Charles Blundell. "Generalization of reinforcement learners with working and episodic memory." Advances in neural information processing systems 32 (2019).

[4] Lampinen, Andrew, Stephanie Chan, Andrea Banino, and Felix Hill. "Towards mental time travel: a hierarchical memory for reinforcement learning agents." Advances in Neural Information Processing Systems 34 (2021): 28182-28195.

[5] Parisotto, Emilio, Francis Song, Jack Rae, Razvan Pascanu, Caglar Gulcehre, Siddhant Jayakumar, Max Jaderberg et al. "Stabilizing transformers for reinforcement learning." In International conference on machine learning, pp. 7487-7498. PMLR, 2020.

**Questions:**

- It is unclear why the inductive bias (composite memory and contextual drift) is designed specifically for RL setting. The bias is very general and can be used for both SL and RL. This raises the questions such as whether FFM can show good performance under SL? Why only test it on RL?
- It is hard to link Eq. (11) to the contextual drift motivation where $\omega$ should be recurrently computed.
- It is also unclear why the decay should be complex-valued. The authors should present the explanation for the changes compared to the original motivation Eq. 8-10.
- The choice of gating for the Cell seems arbitrary. Could you explain the reasons behind these designs, or do you follow any prior papers/ideas?

**Limitations:**

More investigation is needed on the initialization of the decay. Now, it seems to be heuristic and unclear if it still works for other tasks or settings.

---

> ### Author Rebuttal · Authors · 2023-08-05
>
> Thank you for the detailed review. Our primary concern in this paper was to design a __more efficient RNN for model-free RL__, with the idea that anyone with an existing recurrent RL implementation (e.g. CleanRL's recurrent PPO) could easily replace their LSTM/GRU with FFM, and train a comparable policy using fewer resources. We believe we have succeeded in that regard.
> ### Novelty and the Relationship to [1]
> The $\alpha$ term in FFM's aggregator produces similar decay to $\lambda$ from [1], but we disagree with the premise that our approach is a simple modification of [1].
> - We benchmark FFM against modern variants of [1], such as the Fast Autoregressive Transformer (FART) and the Fast Weight Programmer (FWP) (Fig. 3 and Appendix Fig. 10).
> - [1] only studies a single RL task which we would argue is insufficient to draw conclusions on RL performance.
> - [1] does not appear parallelizable over time like FFM, meaning it will be less efficient and on-par with RNN efficiency (Eq. 2 in [1] shows $h_{s+1}$ has a nonlinear dependence on $h_s$)
> - We compute an outer product, but it is not using the recurrent state $h(\tau) h(\tau)^\top$ as in [1], but rather with a function of input the and the context vector $f(x_\tau) \omega(\tau)^\top$ (using their notation).
>
> The temporal context fixes some of the theoretical shortcomings of [1], and is a non-trivial addition that requires numerous additional considerations (see __An Explanation of Temporal Context__ in the __Author Rebuttal__). Let us summarize and inspect the ablations from Table 2:
>
> |Temporal Context ($\omega$)|Score|Std.|
> |-|-|-|
> |None|0.382|0.005|
> |Fixed|0.383|0.005|
> |Learned|0.399|0.005|
>
> We can see that a learned context is important and statistically significant ($p=0.0141$ according to T-test computed via [A]). Furthermore, if the context $\omega$ is not learned, then equations 20-22 fall apart and FFM can no longer approximate any temporal convolution.
>
> The case behind learning decay $\alpha$ is a bit weaker, but still provides a marginal benefit at the cost of 32 additional parameters. Readers are free to use our model with a fixed $\alpha$ if they find the results unconvincing -- it does not diminish our contributions.
> ### MANNs and Transformers
> Many of your cited papers are indeed missing from our related work. Given our paper's scope and focus on efficiency, we will explain why we did not include or compare to them:
> - [2] is not peer-reviewed and exceedingly complex -- there is not one working public implementation after 5 years. Furthermore, the backbone is an RNN which we show is much less efficient than FFM.
> - [3] is not parallelizable over time due to writes and reads to external memory, making it slow like the RNNs we tested. Furthermore, it is not model-free RL as it requires predicting the next observation (i.e., it is a dynamics model).
> - [4] and [5] combine an LSTM/GRU with a transformer, resulting in quadratic space complexity while also being slow to run because it cannot be parallelized over time. We cite [5] and tried to run it, but our GPU did not have enough memory to fit it given the batch sizes used by POPGym.
>
> It is possible that some of these methods could outperform FFM on a per-sample basis, but not on a time or hardware efficiency basis.
>
> Furthermore, the LSTM and GRUs from the POMDP-Baselines experiments used hand-tuned initialization and task-specific hyperparameters from Ni et al. Our FFM model outperformed those models while using the exact same hyperparameters.
> ### Experimental Gaps Between FFM and the GRU
> Fig. 3 shows that on average, we exceed the returns of the best POPGym model (GRU) by ~10%, without overlapping 95th percentile error bounds. Fig. 4 breaks this down further, showing that we perform noticeably worse on 4 tasks and noticeably better on 18. For POMDP-Baselines, FFM outperforms the LSTM/GRU on 12 out of 14 tasks. We argue that this paints a clear picture of FFM outperforming RNNs, even before considering efficiency.
> ### Inductive Biases
> You are correct in that FFM's inductive bias could be useful on SL tasks, and in fact, FFM could be run without modification on SL tasks. Frankly, we are less familiar with the wider SL literature, and as you point out in [1], there are already many good papers on the subject (which we will add to our related work). The POPGym paper shows that there is an unpredictable performance gap between memory in SL and RL, so we argue that RL-focused experiments are especially useful to the RL community.
> ### Recurrent Temporal Context
> The purpose of Eq. 10 is to show that in psychology, context changes gradually. We can reformulate our context implementation recurrently like in Eq. 10 as $\omega_{t+1} = e^{iw} \odot \omega_t$. Please see the section entitled __An Explanation of Temporal Context__ in the __Author Rebuttal__ response for why it is crucial that the temporal context be complex-valued.
> ### Cell Gating
> FFM output gating is very similar to the GRU output gating: $h_t = (1 - z_t) \odot h_{t-1} + z_t \odot g_t; \quad z_t = \sigma(W x_t + b + \dots)$. Accessing the previous state $h_{t-1}$ in FFM would break the parallel nature of FFM, so we replace the $h_{t-1}$ with the input $x_t$, resulting in a skip-connection that improves gradient propagation upstream of the FFM module. It also provides a shortcut for downstream layers to access the current observation.
>
> The input gating approximates the product of the Bernoulli random variable $B$ with input $x$ from Eq. 7-8. We tried sampling from an actual Bernoulli distribution using the reparameterization trick, as well as approximating sparsity via $B_t x_t = ReLU(x_t)$. Our current approximation worked best.
>
> [A] https://www.statskingdom.com/140MeanT2eq.html

---

### Official Review · Reviewer_qKYK · 2023-07-03

**Soundness:** 4 excellent
**Presentation:** 3 good
**Contribution:** 3 good
**Rating:** 8
**Confidence:** 4

**Summary:**

This work introduces a new recurrent neural network layer called Fast and Forgetful Memory (FFM), which, differently from traditional architectures such as GRUs and LSTMs, FFM is explicitly designed for deep reinforcement learning

**Strengths:**

The authors present a novel and promising memory layer for reinforcement learning, the core ideas of the layer are intuitive. The theoretical foundation is well explained and easy to follow, I didn't spot any errors. Also the empirical analysis is extensive. Authors cover time and space complexity of the layer which is very beneficial to get a general idea of its advantages.

In general I think is a good work and a strong accept

**Weaknesses:**

No major weaknesses that I can tell, just a few minor points:
- In the abstract I would not say so broadly that all model-free approaches are using architectures from SL I think that applies to RNNs but mainstream transformers used in RL were specifically redesigned for this framework
- Line 104, I would explain what traces are
- Line 117, remove brackets from citations that are part of the sentence
- Page 4, I would do an effor to inegrate the footnote in the main text, it would make the section much easier to follow

**Questions:**

- Do you foresee any possible issues of instability?
- How long were the temporal dependencies tested in the experiments?

**Limitations:**

I think authors did fine work addressing the limitations of their contribution at the end of the paper

---

> ### Author Rebuttal · Authors · 2023-08-05
>
> Thank you for taking time to review, we hope you enjoyed the paper.
>
> Thank you for the corrections, we will fix these at once. The footnote was originally in the main text, but it took up a lot of space and we were worried it might put some readers to sleep. We will see where we can fit it back in. We briefly cite the Stabilized Transformers for RL paper in the introduction, but we will also add it to the abstract.
>
> As described in the limitations, floating-point instabilities can result from learning very large values of $\alpha$. However, once we clamped $\alpha$, we did not observe any other instabilities unique to FFM. It occasionally overestimated Q values on the off-policy tasks, but not any more often than the LSTM or GRU. As we note in the experiments section, we did not change any hyperparameters for any benchmarks, so FFM should be stable enough for an average user to apply it to standard RL tasks.
>
> The temporal dependencies varied by experiment. On the longer side, the POPGym navigation experiments have a maximum episode length of 1024 timesteps, but it is not clear if the dependencies span all 1024 timesteps. More strictly speaking, the POPGym RepeatFirstHard environment has a temporal dependency of 832 timesteps. We would guess that the POPGym benchmark has an average temporal dependency measured in low hundreds of timesteps, but we do not have a hard number. The POMDP-Baselines benchmark tends to stick to BPTT lengths of 32 or 64, so we suspect it has a lower temporal dependency than POPGym, but we cannot be sure.

---

> > ### Comment · Reviewer_qKYK · 2023-08-18
> > **Rebuttal Response**
> >
> > Thank you for addressing my questions and for the extra effort for the changes, while the paper was quite clear I think they will make the paper even easier to follow for future readers. I keep my strong acceptance recommendation

---

### Official Review · Reviewer_Zkgc · 2023-07-04

**Soundness:** 3 good
**Presentation:** 3 good
**Contribution:** 3 good
**Rating:** 7
**Confidence:** 3

**Summary:**

The paper introduces Fast and Forgetful Memory (FFM), a memory model designed for reinforcement learning. FFM is based on theories of composite memory and contextual drift from computational psychology. It consists of two main components: an aggregator and a cell. The aggregator computes complex-valued summaries of inputs, while the cell performs input gating and extracts real-valued outputs from the summaries. Conceptually, the aggregator implements composite memory by blending individual traces and incorporating an exponential forgetting mechanism. It further enables relative-time reasoning by applying a temporal context. On a suite of POMDP tasks from POPGym and POMDP-Baselines, FFM-enabled recurrent RL algorithms outperform those running on standard RNN. Moreover, since FFM belongs to the class of hybrid memory models, which provides equivalent recurrent and closed-form formulae, it achieves linear space complexity and logarithm time complexity with parallelism. This makes FFM an efficient and performant memory model that can be used as a drop-in replacement for RNN in recurrent RL algorithms.

**Strengths:**

- The paper proposes a novel memory mechanism for reinforcement learning by drawing inspiration from the field of computational psychology. It represents its cell state as a collection of traces and features numerous inductive biases for composition, forgetting, and relative-time reasoning. These inductive biases collectively lead to improved performance on RL tasks.
- The experiments are substantial and thorough, providing ample evidence that FFM can be used as a drop-in replacement for RNN in recurrent RL algorithms. The ablation experiments and analysis provide further insight into the significance of each design choice.
- The contributions of this paper are significant, especially considering the importance of memory models in partially-observable RL.

**Weaknesses:**

- The intuition behind some inductive biases is elusive. Please refer to the Questions section.

**Questions:**

Besides drawing inspiration from computational psychology, are there intuitive or formal reasons that the proposed inductive biases lead to better performance on reinforcement learning tasks compared to the various instantiations of RNN? In particular:
- Why is storing memory as individual traces more desirable than having a single context vector?
- Is there a benefit to using imaginary temporal context over standard position encoding?
- Is there evidence that temporal context enables relative-time reasoning?

**Limitations:**

The authors address the following limitations in their paper:
- Performance decreases with larger context sizes, presumably because of a larger number of periodic functions resulting in more local extrema in optimization.
- Limitation in the number of experiments.
- Repeated multiplication of exponentials can lead to a loss of numerical precision.

---

> ### Author Rebuttal · Authors · 2023-08-06
>
> Thank you for taking the time to review our paper, and thank you for your interest. We hope we are understanding your first question correctly: we only ever store the __sum__ of time-contextualized traces. We assume psychologists like using distinct traces like Eq. 6 because they are conceptually easier to reason over, but from a computational standpoint, they are very inefficient and scale poorly to large sequences. Note that in our paper, every time we say "context" we are explicitly referring to the temporal context (i.e., $e^{iw}$). We can think about the distinct traces being present in the sum, which allows us to do tricks like multiplying by exponentials to "shift" the recurrent state. Please see the section titled __An Explanation of Temporal Context__ in the __Author Rebuttal__ section for why an imaginary context is beneficial.
>
> We do not report it in the paper, but the RepeatPreviousEasy task where the agent must output the observation from $k$ timesteps ago was unsolvable without temporal context. We believe our context-free model had trouble determining which observation it saw precisely $k$ timesteps ago. This extended to other tasks (e.g. Autoencode and others that we do not recall) that depended on the order of observations. Adding context fixed this issue. Fig. 6 provides some evidence for this conclusion.
>
> In Fig.6, FFM empirically learned a context period of $2k = 64$. With this context period, the recurrent state will "spike" at half the period $k=32$. In other words, at $t=32$, the imaginary component of $x_{t-k}$ in the recurrent state will be $i \sin(\omega t) x_{t-k} = x_{t-k}$ and the real component will be $\cos(\omega t) x_{t-k} = 0$. The downstream layers should be able to pick up on this "spike" and output the observation $x_{t-k}$. Put another way -- it is mighty suspicious that the periods our learned context parameters dropped from various periods of $>400$ to precisely $2k$. One might conclude that gradient descent is pushing the context period to the environment period because it is beneficial!

---

> > ### Comment · Reviewer_Zkgc · 2023-08-16
> >
> > Thanks for addressing my questions. I appreciate the clarifications on the traces and the temporal context. The RepeatPreviousEasy result demonstrates that temporal context indeed enables relative-time reasoning, and I suggest including it in the revision. I will keep my positive score and suggest an acceptance.

---

### Official Review · Reviewer_ARGZ · 2023-07-15

**Soundness:** 2 fair
**Presentation:** 2 fair
**Contribution:** 2 fair
**Rating:** 5
**Confidence:** 2

**Summary:**

The paper aims to propose an algorithm-agnostic memory model specifically designed for reinforcement learning (RL). The proposed neural network (NN) model is claimed to offer advantages over existing sequence-based memory models like Recurrent Neural Networks (RNNs) and can be seamlessly substituted for RNNs. The key advantage over RNNs is that the model can process a sequence in parallel, making it more efficient compared to traditional RNNs. The main design components of the model include:
A context drift formula that computes the embedding of sensory information with context.
A decaying mechanism incorporated into the recurrent formula to compute a summary of the embedding.
Gating mechanisms used to extract the final output from the learned summary of the embedding.

**Strengths:**

1. Appropriate architecture may be suitable for a particular type of tasks. I agree that there might be some interesting special NN architecture that is particularly beneficial for RL tasks. Studying RL-specific NN structure is an important and practically useful research direction.

2. The proposed NN architecture seems to be new.

**Weaknesses:**

The main weaknesses are: clarity of presentation, lack of justifications of some design choices, experiments are not persuasive enough.

Clarification questions:

1. What is the difference between subscript t and n? I found it to be confusing in eq(9). What should be the relation between n and t there?

2. eq(4) already defined x_n as the embedding of observation-action pair, why eq(9) redefined this notation?

And what is hat{x}? It is said that it is the raw sensory information, isn’t it part of observation variables?

What is the context variable in RL? Do you simply define omega as a constant one vector?

I strongly suggest the authors to use a particular RL environment to explain all these notations. I found it to be confusing when reading notations in the paper.

Justify some designs:

1. As a new NN architecture,  it is critical to study which component of the proposed methods matters.
At least, the paper should study the necessity of exponential decay mechanism, outer product design (eq 9), etc. One possible baseline in the experiments could be simply adding a decaying mechanism in the hidden state of a regular RNN so that it can somehow show the importance of the decay mechanism.

I have doubts on why decaying is good. Why it makes sense to set up an expiration for each feature unit? In existing literature, it is critical to always maintain a so-called coverage in the state space to learn an optimal policy (please refer to references [1][2][3]). If setting expiration for some feature units, doesn’t it mean manually discard some of the states?

2. The paper should also include discussions with successor features, as the method of computing the hidden state (i.e. eq(7)) is highly similar to how a successor representation is learned. An in-depth study of successor representation may result in better understanding of the proposed architecture.

Experiments.

I found the experiments a bit disappointing. As a general NN architecture, why not also test it on those commonly used RL domains like mujoco, Atari, etc. as mentioned in the abstract, “it is a drop-in replacement for RNN.” There are already existing work using RNN in a variety of RL benchmark tasks. It is better to test the proposed architecture on those well-tested RL domains.

I would like to emphasize that i don’t think it is necessary to restrict the test on POMDP. In function approximation setting, a MDP can be always viewed as a POMDP.

[1] An equivalence between loss functions and non-uniform sampling in experience replay by Scott Fujimoto et al.
[2] Understanding and mitigating the limitations of prioritized experience replay by Pan et al.
[3] regret minimization ER in off-policy RL et al.

**Questions:**

see weaknesses.

---

> ### Author Rebuttal · Authors · 2023-08-05
>
> Thank you for taking the time to review our paper. We believe a reject rating for the issues you raised is unduly harsh, and we believe there might also be some misunderstanding regarding the scope of our paper. The goal of this paper is to design an efficient replacement for RNNs in model-free RL, which summarizes observations from a POMDP into a latent Markov state.
>
> ### Possible Misconceptions
> - There might be some confusion about the word "memory". Your citations [1,2,3] discuss replay buffers for MDPs and do not mention POMDPs or RNNs. More than half of our experiments (POPGym) are on-policy and do not use a replay buffer. We are concerned with "memory" as it applies to an online summary of observations, rather than "memory" in the sense of storing and replaying state-action-reward tuples. We will add a sentence to the introduction to better explain our definition of "memory".
> - Solving MDPs provides no signal on the capacity of memory models to solve POMDPs. This is because MDPs can be solved without memory [A], while POMDPs cannot [B]. The goal of this paper is to develop a memory model to solve POMDPs.
> - We only use pre-existing benchmarks in our experiments. Our experiments from the POMDP study by Ni et al. (Fig. 5) are the MuJoCo tasks you are thinking of (Walker, Ant, HalfCheetah, Hopper), but replicated in the PyBullet engine and modified to be partially observable. We also test on POPGym, which is the largest POMDP benchmark available. Atari is expensive to train recurrent policies on, and we have already used over 50 wall-clock days training. Furthermore, many Atari games are MDPs or can be made MDPs by concatenating the last four observations, which is insufficient to evaluate long-term memory.
> - We perform an extensive ablation of seven different FFM architectures in Table 2, including four decay variants. See the following rebuttal section for more information.
>
> ### The Case for Decay and Other Architectural Decisions
> Decay is already built into most RNNs. LSTMs and GRUs explicitly decay the recurrent state using the forget gate [C]. The forget gate activation $f_t = \sigma(\dots)$ in an LSTM would produce elementwise decay of the recurrent cell state: $c_t = f_t \odot c_{t-1} + \dots$ just like FFM's elementwise decay
> .
> - We ablate decay in Table 2, showing that without decay our model performs worse than the GRU (Fig. 3). Thus, decay is absolutely critical in our model.
> - If forgetting is detrimental to a specific task, our model will learn $\alpha = 0$ through gradient descent, which corresponds to no decay. In Fig. 6, we record the decay rate and show it decreases over time for a long-term memory task.
> - The outer product architecture follows from computational psychology (Eq. 9-10). We tried variants without the outer product (e.g., where each dimension had its own $\alpha, \omega$), but we did not report them.
> - We are not familiar with successor features. Perhaps we could use them to investigate recurrent states produced by memory models, but we already investigate how FFM builds recurrent states in Fig. 6.
>
> ### Notation
> Our notation is different than a normal RL paper because we need to represent both computational psychology and RL using a single notation. That said, we still try to follow standard ML conventions (e.g. $x$ is NN input, $y$ is output, $s$ is the recurrent state, $t,n$ are time/sequence indices, $o$ is observation, etc.). We will do a second pass through all the notation to see if anything can be improved. If you have specific suggestions, we would be happy to make changes.
>
> - In Eq. 9-10, the $t$ is just a placeholder for the index/timestep. We can rewrite Eq. 9-10 replacing $t$ with $n$ if this resolves the issue.
> - We introduce the hat operator in Eq. 9 to make it clear that Eq. 7-8 would store the trace after applying the context ($x$) rather than the observation without context ($\hat{x}$). We will rewrite equations 9 and 10 to be more explicit.
> - The context variable is $\exp(i\omega)$, which is combined with the decay term and defined in Eq. 12. The $\omega$ term is a learned weight of size $c$.
>
> [A] Sutton, Barto. _Reinforcement Learning: An Introduction._ Section 4.4 (http://incompleteideas.net/book/ebook/node44.html)
>
> [B] https://cs.brown.edu/research/ai/pomdp/tutorial/pomdp-background.html
>
> [C] https://en.wikipedia.org/wiki/Long_short-term_memory

---

> > ### Comment · Reviewer_ARGZ · 2023-08-13
> >
> > Thank you for your reply. It definitely clarifies things.
> >
> > Still have some doubts.
> >
> > 1. regarding the scope. I understand that you focus on POMDP. What I mean is, under function approximation setting, you can always consider some information from the original state variables that get lost after approximation, as a result, testing on those traditional MDPs still makes sense. We should expect that at least the proposed method is not worse than a regular RNN. This would significantly strengthen the paper and the generality of the proposed method.
> >
> > 2. "The outer product architecture follows from computational psychology (Eq. 9-10). We tried variants without the outer product (e.g., where each dimension had its own), but we did not report them."
> >
> > Borrowing results from psychology does not direct indicate its utility in AI/ML. Outer product is not a cheap operation (computation & memory), so I think justifying such choice is not trivial.
> >
> > I will go through other reviews carefully and may update my score later.

---

> > > ### Author Response · Authors · 2023-08-14
> > >
> > > Thank you for the quick response.
> > >
> > > ### 1.
> > > Let us make sure that we understand the premise. Are you saying that if an MLP for some reason ignores parts of the state (e.g., certain joint angles in MuJoCo), we can recover them using memory? We argue that it is much easier to recover "lost" inputs through gradient descent on the MLP than with gradient descent on memory. A suitably large MLP can approximate the optimal value function to arbitrary precision according to the universal approximation theorem.
> > >
> > > Fig. 5 shows how our model performs on the standard MuJoCo tasks if we obscure the agent's velocities/positions, forcing the memory model to infer position from the velocity or infer velocity from the position. But these are POMDPs, not MDPs. We would actively discourage anyone from using our model on MDPs.
> > >
> > > ### 2.
> > > You are right that psychology alone does not indicate utility in ML, but we demonstrate the utility of our method by outperforming all the baselines. The outer product is heavily used in ML today, most notably by the transformer which takes the outer product of keys and queries to compute the attention matrix. The FART baseline from POPGym also takes an outer product.
> > >
> > > As we show in Fig. 3, our approach is already cheaper in both memory and time than a variety of methods (including the GRU), given equivalent recurrent state sizes. Given large input vectors, an outer product can consume a lot of memory, but FFM is very efficient with regard to recurrent state size (the largest outer product we compute in this paper is 32 x 8). Even on my laptop CPU, the outer product is fairly quick:
> > > ```python
> > > In [1]: import torch
> > > In [2]: a = torch.rand(128, 32)
> > > In [3]: b = torch.rand(128, 8)
> > > In [4]: %timeit torch.einsum('bi, bj -> bij', a, b)
> > > 21.7 µs ± 16.1 ns per loop (mean ± std. dev. of 7 runs, 10,000 loops each)
> > >
> > > In [5]: d = torch.rand(1, 8)
> > > In [6]: c = torch.rand(32, 1)
> > > In [7]: %timeit c @ d
> > > 737 ns ± 0.299 ns per loop (mean ± std. dev. of 7 runs, 1,000,000 loops each)
> > > ```
> > > During training and inference, the cost of the outer product is negligible compared to the cost of the linear layers in our model.

---

> > > > ### Author Response · Authors · 2023-08-17
> > > >
> > > > We repurposed the POPGym PPO framework to run additional experiments on various MuJoCo MDPs. We do not have time to tune hyperparameters, so we use the hyperparameters from the POPGym paper for all tasks. We compare FFM against the GRU, and run three trials for each model/task combination.
> > > >
> > > > | Task          | Model   |Return  | Std. Dev.  |
> > > > |---------------|---------|--------|------------|
> > > > | Reacher-v4    | GRU     |-5.0    |  0.4      |
> > > > |               |__FFM__ |__-4.6__|  0.9      |
> > > > | Hopper-v4     | GRU     |1064    |644         |
> > > > |               |__FFM__ |__2454__|258         |
> > > > | HalfCheetah-v4|__GRU__ |__1872__|504         |
> > > > |               | FFM     |1580    |386         |
> > > > | Swimmer-v4    | GRU     |136     | 0.8        |
> > > > |               |__FFM__ |__138__ | 6         |
> > > > | Walker2d-v4   | GRU     |  1531  | 448        |
> > > > |               |__FFM__ |__2588__|  749       |
> > > >
> > > >
> > > > Please note that FFM was not designed for MDPs, so we do not expect state of the art performance on these benchmarks. FFM performs noticably better on the Hopper and Walker2d tasks, while the GRU performs slightly better on HalfCheetah. The rest are quite close.
> > > >
> > > > We are also currently running an ablation on the outer product, comparing it with a previous version of FFM that uses the elementwise product instead. In this version, each dimension of the recurrent state has a separate $\alpha, \beta$. We will add the results to Table 2 when finished. This experiment consists of 135 runs (45 tasks, three trials each) and may not finish before the discussion deadline. We previously found this version to perform worse than the outer product variant, so there should be no surprises.
> > > >
> > > > If you find that these results address your concerns, please consider updating your review.

---

> > > > > ### Comment · Reviewer_ARGZ · 2023-08-17
> > > > >
> > > > > Dear authors, thank you for your clarification and for your efforts in enhancing the experiments. I look forward to seeing these results incorporated into the next version.

---

### Author Rebuttal · Authors · 2023-08-06

Thank you to all the reviewers and ACs who spent time reviewing this paper. We are happy to hear that our proposed method is "novel and promising", as well as "inuitive...well explained, and easy to follow", that "the empirical analysis is extensive" and "The contributions of this paper are significant, especially considering the importance of memory models in partially-observable RL."

We believe some readers might benefit from a better explanation of temporal context, so we will move the subsection "Efficient Convolution with Infinite-Extent Filters" to the appendix to make room for an expanded explanation of temporal context. In the meantime, let us explain the rationale behind our implementation of temporal context.

### An Explanation of Temporal Context
The context is traditionally real-valued in computational psychology (Eq. 9-10), but our implementation leverages the complex domain to break symmetries and improve relative-time reasoning. Since we are effectively summing inputs to compute the recurrent state, we can run into ordering symmetries in the real domain because addition is commutative.

Let us begin with a simple example very similar to FFM aggregation, but without context and of a single dimension. Our goal is to compute a recurrent state $s_n$

$$ s_n = x_1 e^{(n-1)\alpha} + x_2 e^{(n-2)\alpha} + \dots $$

where $\alpha$ is the decay term. Consider the case where we learn a real decay term of $\alpha = 0$ (no decay), perhaps because two inputs $x_j, x_k$ are very important and cannot be forgotten. For clarity and without loss of generality, assume all $x \notin \{x_j, x_k\}$ are worthless and we learn to map them to zero. Then, our recurrent state is

$$ s_n = e^{(n - j) \alpha} x_j + e^{(n - k) \alpha} x_k = x_j + x_k. $$

How are we to know whether we observed $x_j$ first or $x_k$ first, or how long ago they occurred? In the current case, we cannot know either. We could add a transformer positional encoding, but this would not solve the ordering ambiguity, as we would have

$$ s_n = x_j + \textrm{pos}(j) + x_k + \textrm{pos}(k) = x_j + \textrm{pos}(k) + x_k + \textrm{pos}(j). $$

Using an imaginary context $e^{i\omega}$ breaks these symmetries. With imaginary context, our recurrent state becomes

$$ s_n = x_j e^{(n - j) i\omega} + x_k e^{(n - k) i\omega}$$

Now, we can tell whether $x_j$ or $x_k$ came first, and how old each observation is. Thanks to Euler's Formula ($e^{i\omega} = i \sin \omega + cos \omega$), we can shift these inputs infinitely far into the past or future without losing the relative distance between $j$ and $k$. In other words, as $n \to \infty$, the phase shift between $e^{(n - j) i\omega}$ and $e^{(n - k) i\omega}$ remains constant.

Let us demonstrate how we "shift" the recurrent state with a concrete example. At timestep $n+1$, we shift the recurrent state one step into the past via multiplication:

$$ s_{n+1} = e^{iw} (x_j e^{(n - j) i\omega} + x_k e^{(n - k) i\omega}) = x_j e^{(n + 1 - j) i\omega} + x_k e^{(n + 1 - k) i\omega}. $$

In practice, this shift requires computing very large exponents and quadratic memory complexity. Appendix C derives tricks to avoid these issues. The use of complex exponentials also makes FFM able to approximate any convolutional filter (subsection Efficient Convolution with Infinite-Extent Filters), providing a strong theoretical reason for learning the context parameter $\omega$.

---

### Decision · Program_Chairs · 2023-09-21

**Decision:**

Accept (poster)

**Comment:**

I am thrilled to convey my recommendation for the acceptance of your paper to be presented at NeurIPS.

This is effectively a new neural net architecture drop-in replacement for RNN's. It's well written and appears novel enough. It could have gone either way. Nothing really exciting, nor dramatically disappointing.

NeurIPS is a prestigious platform that attracts the brightest minds in the field, and your paper's acceptance adds to the conference's reputation for excellence. I am genuinely excited to see your work presented and discussed among peers who share your passion for pushing the boundaries of knowledge.